# Indication of high basal melting at the EastGRIP drill site on the Northeast Greenland Ice Stream

Ole Zeising[1,2] and Angelika Humbert[1,2]

[1]Alfred-Wegener-Institut Helmholtz-Zentrum für Polar- und Meeresforschung, Bremerhaven, Germany
[2]University of Bremen, Department of Geosciences, Germany

**Correspondence:** Ole Zeising (ole.zeising@awi.de)

**Abstract.** The accelerated ice flow of ice streams that reach far into the interior of the ice sheets, is associated with lubrication of the ice sheet base by basal melt water. However, the amount of basal melting under the large ice streams – such as the Northeast Greenland Ice Stream (NEGIS) – is largely unknown. In-situ measurements of basal melt rates are important from various perspectives as they indicate the heat budget, the hydrological regime and the relative importance of sliding in glacier motion. The few previous estimates of basal melt rates in the NEGIS region were $0.1\,\mathrm{m\,a^{-1}}$ and more, based on radiostratigraphy methods. These findings raised the question of the heat source, since even an increased geothermal heat flux could not deliver the necessary amount of heat. Here, we present basal melt rates at the recent deep drill site EastGRIP, located in the center of NEGIS. Within two subsequent years, we found basal melt rates of $0.19 \pm 0.04\,\mathrm{m\,a^{-1}}$ that are based on analysis of repeated phase-sensitive radar measurements. In order to quantify the contribution of processes that contribute to melting, we carried out an assessment of the energy balance at the interface and found the subglacial water system to play a key role in facilitating such high melt rates.

## 1 Introduction

Ice sheet models are used to quantify the contribution of the Greenland Ice Sheet (GrIS) to future sea-level rise under different climatic scenarios. In these simulations, the distinctive extent of Greenland's largest ice stream – the Northeast Greenland Ice Stream (NEGIS, Fig. 1) – can only be reproduced well if a higher-order approximation is considered for the momentum balance and initial states are based on inversion (Goelzer et al., 2018) or involve subglacial hydrological models (Smith-Johnsen et al., 2020a). Primarily, this is due to the model's inability to accurately represent lubrication and thus the subsequent sliding at the ice stream base that occurs.

The NEGIS is the only large ice stream in Greenland, extending from a distance of 100 km from the ice divide over a length of about 700 km towards the coast (Fahnestock et al., 1993, 2001b; Joughin et al., 2001). It drains about 12 % of Greenland's ice through three major outlet glaciers Nioghalvfjerdsbrae, Zachariæ Isstrøm and Storstrømmen Glacier (Rignot and Mouginot, 2012). Loss of the floating tongue of Zachariæ Isstrøm has already led to ice flow acceleration and increased mass loss (Mouginot et al., 2015). Consequently, it is expected and projected that NEGIS will contribute significantly to sea-

level rise in the future (Khan et al., 2014), highlighting the importance of understanding the general ice flow dynamics and its driving mechanisms.

One hypothesis for the genesis of NEGIS is locally increased basal melting at the onset area that enables and enhances basal sliding (Fahnestock et al., 2001a; Christianson et al., 2014; Franke et al., 2021) and forms a subglacial hydrological system. The coupling with basal sliding is facilitated via the water pressure, so that the sliding velocity rises with increasing water pressure (e.g., Beyer et al., 2018; Smith-Johnsen et al., 2020a). However, little is known about the amount of subglacial water below the up to $\sim 3300\,\mathrm{m}$ thick ice sheet.

First estimates of basal melt rates by Fahnestock et al. (2001a) and later by Keisling et al. (2014) and MacGregor et al. (2016) are based on the interpretation of chronology in radiostratigraphy. All three studies found melt rates of $0.1\,\mathrm{m\,a^{-1}}$ and more – which is extremely large for inland ice. However, these estimates may be prone to limited validity given the assumptions about the flow regime and constant accumulation rate. The cause for such intensive melt was attributed to a high geothermal heat flux which possibly originates from the passage of Greenland over the Icelandic hot spot (Fahnestock et al., 2001a; Rogozhina et al., 2016; Martos et al., 2018; Alley et al., 2019).

In order to directly observe, among other things, flow regimes and basal conditions of ice streams, an ice core is being drilled as part of the East Greenland Ice-Core Project (EastGRIP) near the onset of the NEGIS. Here, surface velocities reach about $57\,\mathrm{m\,a^{-1}}$ (Hvidberg et al., 2020) and the NEGIS widens (Fig. 1). Smith-Johnsen et al. (2020a) forced an ice model with a locally increased heat flux below the EastGRIP drill site. They found that a heat flux of $0.97\,\mathrm{W\,m^{-2}}$ (corresponding to a basal melt rate of $0.1\,\mathrm{m\,a^{-1}}$ (Fahnestock et al., 2001a)) is necessary to reasonably reproduce the velocities of NEGIS. By utilizing a coupled subglacial hydrology and ice sheet model, Smith-Johnsen et al. (2020b) demonstrated the large impact of an uncertainty in geothermal heat flux on the flow of NEGIS arising from the subglacial hydrological system, hence from basal melting and water pressure, as well as from friction.

However, measurements with an adequate accuracy are still required to narrow down the basal melt rates further. Here, we present the first estimates of basal melt rates from repeated in-situ phase-sensitive radar measurements from the EastGRIP drill site and consider the contribution of different heat sources at the ice base.

## 2 Data and Methods

### 2.1 Instrument, data acquisition and processing

The autonomous phase-sensitive radio-echo sounder (ApRES; Brennan et al., 2014; Nicholls et al., 2015) is a low-power, ground-based radar that allows measurements to be carried out autonomously with a selected interval over long periods of time. By analyzing the phase shifts of the return signals, the vertical displacements of internal reflections and of the base can be precisely determined in millimeter range. Thus, the ApRES is often used to determine Lagrangian basal melt rates and their temporal variability of ice shelves (e.g., Stewart et al., 2019; Washam et al., 2019; Vaňková et al., 2020). In order to derive an annual mean basal melt rate at the EastGRIP drill site, we deployed an ApRES within a near surface trench (Fig. 1). The ApRES performed a measurement once a day during winter from 08/2017 – 04/2018 and 08/2018 – 05/2019.

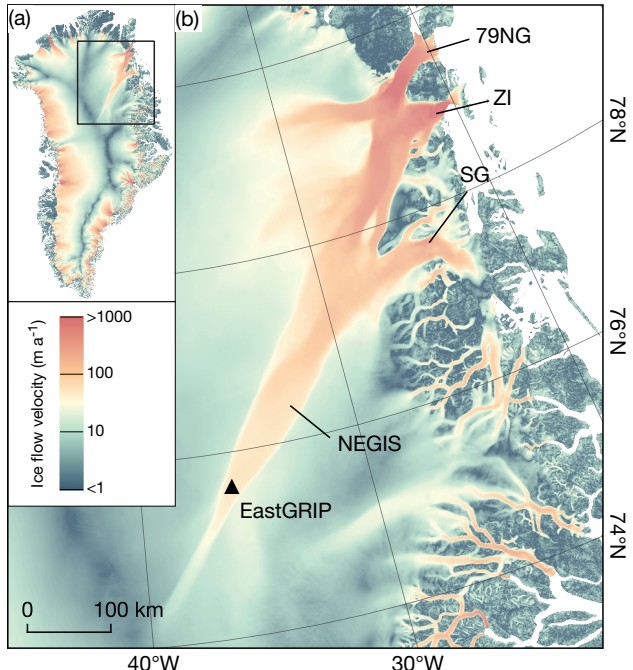

**Figure 1.** Surface ice flow velocity map of the Greenland Ice Sheet (Joughin et al., 2018). The box in the overview map (upper left corner) marks the boundaries of the main figure showing northeast Greenland and the Northeast Greenland Ice Stream (NEGIS), which drains into the three major outlet glaciers, namely Nioghalvfjerdsbrae (79°N Glacier, 79NG), Zachariæ Isstrøm (ZI) and Storstrømmen Glacier (SG). The location of the EastGRIP drill site is denoted by the black triangle.

In the following, we shortly describe the theory of operation of the ApRES and the signal processing, whereas a more detailed description is given by Brennan et al. (2014); Nicholls et al. (2015) and Stewart et al. (2019). Within a single measurement, the ApRES transmits a sequence of 100 chirps, each with a duration of 1 s in which the frequency of the transmitted electromagnetic wave is increased from 200 to 400 MHz. After reflection, the received signal is mixed with a replica of the transmitted signal and sampled with 40 kHz (Nicholls et al., 2015). Since the frequencies of the resulting deramped signal are related to the two-way travel time, a spectral analysis needs to be done in the processing to obtain depth profiles of the amplitude and phase. For the conversion from travel time to depth, we used a vertical propagation velocity of $168,914 \, \mathrm{km \, s^{-1}}$ according to the relative permittivity of $\varepsilon_r = 3.15$ (Fujita et al., 2000). Prior to the spectral analysis, the performed chirps were averaged to increase the signal-to-noise ratio. However, due to weak reflections in the lower part of the ice, the signal-to-noise ratio is reduced. As a consequence, no reliable analysis of the data is possible below the noise level depth limit of $\sim 1450 \, \mathrm{m}$, with the exception of the basal return. Next, we present how we derive basal melt rates from the radar data.

## 2.2 Ice thickness evolution

The method we use to derive a basal melt rate is based on the ice thickness evolution equation that is valid in both, the Eulerian and Lagrangian reference system

$$\frac{\partial H}{\partial t} = -\text{div}\,\boldsymbol{Q} + a_s - a_b \tag{1}$$

with the ice thickness $H$, the time $t$, the volume flux $\boldsymbol{Q}$, the surface mass balance $a_s$ and the basal melt rate $a_b$ (positive for melting) (e.g., Greve and Blatter, 2009). Equation (1) states that a temporal change in ice thickness is caused by a changing volume flux arising from deformation and accumulation or ablation at the ice surface and base. It is worth to note that a basal melt rate larger than the accumulation rate only leads to thinning of the glacier, if the volume flux cannot supply sufficient amount of ice to balance this out. The volume flux $\boldsymbol{Q}$ is defined as the vertically integrated horizontal velocities $v_x, v_y(x,y,z,t)$

$$\boldsymbol{Q} = \begin{pmatrix} Q_x \\ Q_y \end{pmatrix} = \begin{pmatrix} \int_0^H v_x \,\mathrm{d}z \\ \int_0^H v_y \,\mathrm{d}z \end{pmatrix}. \tag{2}$$

(Greve and Blatter, 2009) and represents the ice thickness change due to deformation and sliding, thus stretching or compression in the horizontal direction. This may, for example, be due to changes in basal velocities or ice creeping across a bedrock undulation. Using the continuity equation for incompressible materials, $\text{div}\,v = 0$, and Leibniz's integral rule we can rewrite $\text{div}\,\boldsymbol{Q}$ as

$$\text{div}\,\boldsymbol{Q} = \frac{\partial}{\partial x}\int_0^H v_x \,\mathrm{d}z + \frac{\partial}{\partial y}\int_0^H v_y \,\mathrm{d}z = \int_0^H \frac{\partial v_x}{\partial x} + \frac{\partial v_y}{\partial y}\,\mathrm{d}z = \int_0^H \frac{\partial v_z}{\partial z}\,\mathrm{d}z = \int_0^H \dot{\varepsilon}_{zz}\,\mathrm{d}z \tag{3}$$

with $\dot{\varepsilon}_{zz}$ the vertical strain rate $\dot{\varepsilon}_{zz} = \partial v_z/\partial z$.

The recorded ApRES time series allows for a precise estimation of changes in ice thickness $\Delta H$ from the vertical displacement of the basal reflector and of internal layers from consecutive measurements. However, applying the ice thickness evolution equation (Eq. (1)) to the ApRES measurements requires some modifications. Since the ApRES is located within a trench below the surface, the 'measured ice thickness' $H$ is defined as the range between the ApRES and the ice base. The total ice thickness – the range from the surface to the ice base – is about 7 to 8 m thicker and includes the upper firn and snow layers. Thus, $\Delta H$ is independent on the surface mass balance, $a_s = 0\,\mathrm{m\,a^{-1}}$, but influenced by firn densification that significantly affects the vertical displacement in the upper $\sim 100\,\mathrm{m}$. As this is not considered in Eq. (1), we add the term $\Delta H_f/\Delta t$ to correct for the densification process below the ApRES.

Equation (3) states that the divergence of the volume flux in Eq. (1) can be expressed by the depth integrated vertical strain rate. However, we derive vertical displacements $u_z$ from ApRES measurements instead of vertical velocities $v_z$. Thus, we can calculate strain $\varepsilon_{zz} = \partial u_z/\partial z$ for a time period of $\Delta t$. Therefore, Eq. (3) needs to be reformulated as

$$\int_0^H \dot{\varepsilon}_{zz}\,\mathrm{d}z = \frac{1}{\Delta t}\int_0^H \varepsilon_{zz}\,\mathrm{d}z = \frac{\Delta H_\varepsilon}{\Delta t} \tag{4}$$

with the change in ice thickness due to vertical strain $\Delta H_\varepsilon$. Finally, the modified ice thickness evolution equation can be written as

$$\frac{\Delta H}{\Delta t} = \frac{\Delta H_f}{\Delta t} + \frac{\Delta H_\varepsilon}{\Delta t} - a_b \, . \tag{5}$$

All three quantities $\Delta H$, $\Delta H_f$ and $\Delta H_\varepsilon$, which are needed to derive $a_b$, are described by vertical displacements and hence by the radar measurement itself in a consistent manner.

## 2.3  Derivation of basal melt rates

In order to derive vertical displacements of internal layers and of the basal return from the ApRES time series, we slightly modified the processing of Vaňková et al. (2020) (details below). Both methods are based on phase differences estimated from
cross-correlation of the repeated measurements.

Firstly, we divided the depth profile into 6 m segments with a 3 m overlap from a depth of 20 m below the antennas to 20 m above the ice base and a wider segment of 10 m (-9 to +1 m) around the basal return, characterized by a strong increase in amplitude. In order to derive vertical displacements, each depth segment of the first measurement ($t_1$) was cross-correlated with the same segment of each repeated measurement ($t_i$). This is in contrast to Vaňková et al. (2020), who derived displacements
from pairwise time-consecutive measurements ($t_{i-1} - t_i$). The lag of the minimum mean phase difference obtained from the cross-correlation gives the cumulative displacement at the given depth. The range of expected lag was limited by the estimation to the previous measurement ($t_1 - t_{i-1}$). This results in a time series of displacements for each segment individually. The vertical displacement of the basal segment is the change in the measured ice thickness $\Delta H$.

Next, we estimate the vertical strain $\varepsilon_{zz}^{\mathrm{obs}}$ and quantify $\Delta H_f$ as well as $\Delta H_\varepsilon$ based on a regression analysis of the vertical
displacements. To avoid influences of firn densification on the determination of $\varepsilon_{zz}^{\mathrm{obs}}$, we excluded all segments above a depth of 250 m ($\sim 9\,\%$ of all segments). In addition, segments below the noise-level depth limit of $h \approx 1450\,\mathrm{m}$ (where noise prevents an unambiguous estimation) were excluded ($\sim 45\,\%$ of all segments). Furthermore, outliers were filtered out ($\sim 7\,\%$). We found a linear fit $u_z(z)$ of

$$u_z(z) = \varepsilon_{zz}^{\mathrm{obs}} \cdot z + \Delta H_f, \quad 250\,\mathrm{m} \leq z \leq h \tag{6}$$

that best matches the cumulative vertical displacements of the remaining $\sim 400$ segments within the ice. The gradient of this fit is $\varepsilon_{zz}^{\mathrm{obs}}$ and the shift between the intercept at the depth of the ApRES and $\Delta H$ is $\Delta H_f$. However, $\varepsilon_{zz}$ for $z \geq h$ is unknown. Here, we used two scenarios to estimate $\Delta H_\varepsilon$ (Fig. 2, Appendix Fig. A1). First, we assumed that $\varepsilon_{zz}$ is constant with depth:

$$\varepsilon_{zz}^{\mathrm{const}}(z) = \varepsilon_{zz}^{\mathrm{obs}}, \quad 0 \leq z \leq H \tag{7}$$

As a second scenario, we used a vertical strain distribution ($\varepsilon_{zz}^{\mathrm{sim}}$) obtained from an ice sheet model based on inverse surface
flow velocities (Rückamp et al., 2020). Here, $\varepsilon_{zz}^{\mathrm{sim}}$ increases with depth and reaches values of roughly twice $\varepsilon_{zz}^{\mathrm{obs}}$ at the base.

In order to be less dependent on a single measurement, we compute for each of the last 65 days (records; roughly $25\,\%$ of the measurements) of a year an annual melt rate and compute from these 65 melt rate estimates a mean annual value by averaging.

Finally, $\Delta H_\varepsilon$ was derived from Eq. (4) for the two vertical strain distributions ($\Delta H_\varepsilon^{\mathrm{const}}$, $\Delta H_\varepsilon^{\mathrm{sim}}$), and the basal melt rate $a_{\mathrm{b}}$ from Eq. (5). Given errors are based on the standard deviation of the estimates based on the considered 65 measurements and
a $1\,\%$ uncertainty in the signal propagation speed in ice (Fujita et al., 2000). For visualization, we calculated the cumulative vertical displacement referenced to the ice base (Fig. 2).

## 3   Results

The analysis of the 2017/18 ApRES time series revealed a measured ice thickness (distance between radar and ice base) of roughly 2668 m at the EastGRIP drill site with an annual mean change of $-0.471\,\mathrm{m\,a^{-1}}$ (Table 1, Fig. 2). The firn densification
– the intercept (see Fig. 2) of the linear fit at $z = 0\,\mathrm{m}$ (the elevation of the ApRES) – occurring below the radar is $0.074\,\mathrm{m\,a^{-1}}$. We derived a vertical strain of $\varepsilon_{zz} = -0.068 \times 10^{-3}$ from reliable estimates of vertical displacements feasible to a depth of $1450\,\mathrm{m}$. The dynamic thinning of the ice derived from the two scenarios ranges from $-0.181\,\mathrm{m\,a^{-1}}$ ($\Delta H_\varepsilon^{\mathrm{const}}$) to $-0.194\,\mathrm{m\,a^{-1}}$ ($\Delta H_\varepsilon^{\mathrm{sim}}$). This results in a basal melt rate of $0.210 \pm 0.015\,\mathrm{m\,a^{-1}}$. The numbers derived from the time series recorded in 2018/19 differs slightly (Table 1, Appendix Fig. A2). The annual mean change in measured ice thickness is $27\,\mathrm{mm}$ (or $6\,\%$)
lower and the firn densification $15\,\mathrm{mm}$ (or $20\,\%$) larger compared to the values derived in 2017/18. The resulting basal melt rate of $0.167 \pm 0.018\,\mathrm{m\,a^{-1}}$ is $\sim 20\,\%$ lower than the year before. Finally, we derive an averaged melt rate over both years of $0.19 \pm 0.04\,\mathrm{m\,a^{-1}}$.

**Table 1.** Results for measured ice thickness change ($\Delta H$), firn densification ($\Delta H_f$), vertical strain ($\varepsilon_{zz}$), dynamic ice thickness change obtained from a constant vertical strain ($\Delta H_\varepsilon^{\mathrm{const}}$) and a simulation ($\Delta H_\varepsilon^{\mathrm{sim}}$) and basal melting ($a_{\mathrm{b}}$) for both time series projected to 365 days. Negative values contribute to the thinning of the ice column, whereas a positive melt rate represents melting.

| Year | $\Delta H$ (m) | $\Delta H_f$ (m) | $\varepsilon_{zz}$ ($\times 10^{-3}$) | $\Delta H_\varepsilon^{\mathrm{const}}$ (m) | $\Delta H_\varepsilon^{\mathrm{sim}}$ (m) | $a_{\mathrm{b}}$ ($\mathrm{m\,a^{-1}}$) |
|---|---|---|---|---|---|---|
| 2017/18 | $-0.471 \pm 0.008$ | $-0.074 \pm 0.001$ | $-0.068 \pm 0.001$ | $-0.181 \pm 0.001$ | $-0.194 \pm 0.001$ | $0.210 \pm 0.015$ |
| 2018/19 | $-0.444 \pm 0.006$ | $-0.089 \pm 0.002$ | $-0.068 \pm 0.002$ | $-0.182 \pm 0.005$ | $-0.195 \pm 0.005$ | $0.167 \pm 0.018$ |

## 4   Discussion

We used estimated vertical displacements from the upper half of the ice column to estimate the dynamic thinning, since noise
prevents an unambiguous estimation of the vertical strain for the lower half. To cover a range of variations in the dynamic thinning, we used two different scenarios for vertical strain distribution. The resulting dynamic thinning of the simulated vertical strain and the constant strain differs only slightly. However, we cannot exclude the possibility that larger strain values are reached at the base, which would lead to an overestimation of the basal melt rates. In case of a non-existing melt rate, the dynamic thinning of the lower half of the ice column would be, on average, more than four times as large as the one of the upper
half. However, a strong increase is not found in higher-order ice sheet simulations (Rückamp et al., 2020). A frequently used

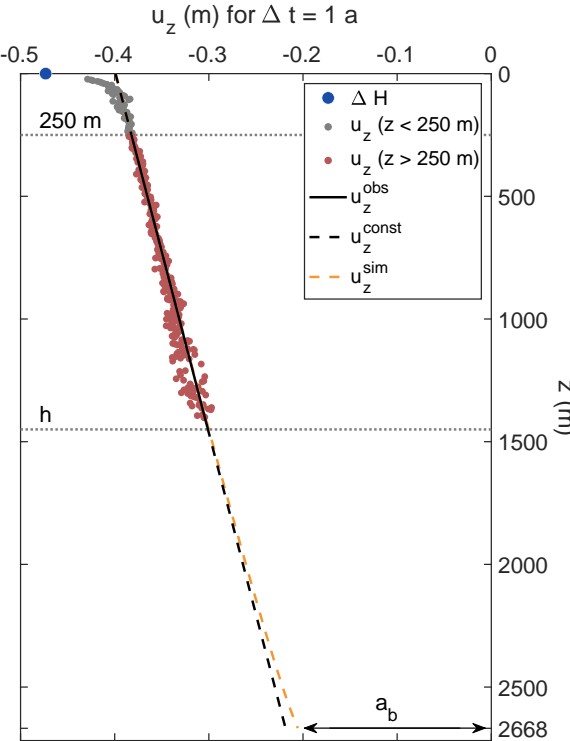

**Figure 2.** Derived vertical displacements $u_z$ of the depth of the ApRES ($\Delta H$; blue dot) and of selected internal layers referenced to the ice base from 2017/18 ApRES time series. Derived displacements used for melt rate estimations are marked by red dots and of layers within the firn by gray dots. The estimated displacements between a depth of $250\,\mathrm{m}$ and $h$ are used to calculate a linear fit (solid black line), the gradient of which is the vertical strain. Extrapolations to the bottom are shown by the dashed lines. The offset at the ice base is caused by basal melting and the difference between the intercept of the linear fit at $z = 0\,\mathrm{m}$ and $\Delta H$ is the firn compaction.

strain distribution (e.g., Fahnestock et al., 2001a; Keisling et al., 2014; MacGregor et al., 2016) that takes into account deviating strain within a shear zone is the Dansgaard–Johnsen distribution model (Dansgaard and Johnsen, 1969). As this model assumes a linearly decreasing strain in the shear zone that reaches zero at the ice base, the resulting basal melt rate at EastGRIP would be even larger. However, the Dansgaard–Johnsen model represents a no-slip boundary condition at the ice base. As this is an

unrealistic assumption in an ice stream, we did not consider the Dansgaard–Johnsen model further. The derived vertical strain is based on more than 300 vertical displacements estimated between the firn-ice transition and about $1450\,\mathrm{m}$. In contrast, the estimation of the displacement of the basal return is based on the phase shift of only one segment around the basal return, slightly above the noise-level. This makes the determination more prone to errors. Instead of comparing the first measurement ($t_1$) with all repeated measurements ($t_i$), the pairwise comparison of time-consecutive measurements ($t_{i-1}$ and $t_i$), as it is

shown by Vaňková et al. (2020), leads to a lower thinning rate of $\Delta H$ in 2017/18 than in 2018/19 ($-0.441 \pm 0.004\,\mathrm{m\,a^{-1}}$ in

2017/18, $-0.467 \pm 0.009\,\mathrm{m\,a^{-1}}$ in 2018/19). Thus, the variability found is not necessarily a variability of the ice sheet system but can rather be influenced by the methodology.

A variation in the selected depth limit of densification, to exclude segments affected by densification, causes slight changes in vertical strain and thus in basal melt rate in the order of millimeters. However, we observed an increased densification rate within the considered 65 records. The increased densification can possibly be a result of increased load from the camp at the surface.

Our derived basal melt rate of $0.19 \pm 0.04\,\mathrm{m\,a^{-1}}$ is above previous estimates from airborne radar measurements. Fahnestock et al. (2001a) and MacGregor et al. (2016) found melt rates in the order of $0.1\,\mathrm{m\,a^{-1}}$ in the vicinity of the EastGRIP drill site, but larger melt rates of $> 0.15\,\mathrm{m\,a^{-1}}$ further upstream in the onset region of NEGIS. Both studies used a constant vertical strain over depth where basal melting occurs. Smith-Johnsen et al. (2020a) found that basal melt rates of $0.1\,\mathrm{m\,a^{-1}}$, derived from a heat flux of $0.97\,\mathrm{W\,m^{-2}}$, are needed at the location of the EastGRIP drill site to reproduce the NEGIS in an ice sheet model.

## 4.1 Considerations of the energy balance at the ice base

In order to constrain the heat flux required to sustain the basal melt rates $a_\mathrm{b}$ derived in this study, we consider the energy balance at the ice base. As for any surface across which a physical quantity may not be continuous, a jump condition is formulated. In typical continuum mechanical formulation, the jump ($[\![\psi]\!]$) of a quantity $\psi$ is defined as $[\![\psi]\!] = \psi^+ - \psi^-$, meaning the difference in the quantity $\psi$ across the interface (Greve and Blatter, 2009). The jump condition of the energy at the ice base reads as

$$[\![\boldsymbol{q}\cdot\boldsymbol{n}]\!] - [\![\boldsymbol{v}\cdot\mathbf{t}\cdot\boldsymbol{n}]\!] + [\![\rho^\mathrm{i}\left(u + \tfrac{1}{2}\boldsymbol{v}^2\right)\left(\boldsymbol{v}-\boldsymbol{w}\right)\cdot\boldsymbol{n}]\!] = [\![\boldsymbol{q}\cdot\boldsymbol{n}]\!] - [\![\boldsymbol{v}\cdot\mathbf{t}\cdot\boldsymbol{n}]\!] + \rho^\mathrm{i} a_\mathrm{b} [\![u]\!] = 0 \tag{8}$$

with the heat flux $\boldsymbol{q}$, the velocity $\boldsymbol{v}$, the velocity of the singular surface $\boldsymbol{w}$, the normal vector $\boldsymbol{n}$ pointing outwards of the ice body, the Cauchy stress $\mathbf{t}$, the ice density $\rho^\mathrm{i}$ and the internal energy $u$ (Greve and Blatter, 2009). The jump of the heat flux $[\![\boldsymbol{q}\cdot\boldsymbol{n}]\!]$ becomes $(\boldsymbol{q}^\mathrm{geo} + \boldsymbol{q}^\mathrm{sw})\cdot\boldsymbol{n} - \kappa(T)\,\mathrm{grad}\,T$, with $\boldsymbol{q}^\mathrm{geo}$ the geothermal heat flux and $\boldsymbol{q}^\mathrm{sw}$ the heat flux from subglacial water with a temperature above pressure melting point, $T$ temperature and $\kappa$ thermal conductivity. For the jump in work of surface forces we find

$$[\![\boldsymbol{v}\cdot\mathbf{t}\cdot\boldsymbol{n}]\!] = \boldsymbol{v}^\mathrm{sw}\cdot\mathbf{t}^\mathrm{sw}\cdot\boldsymbol{n} - \boldsymbol{v}^\mathrm{i}_\mathrm{b}\cdot\mathbf{t}^\mathrm{i}\cdot\boldsymbol{n} \tag{9}$$

with $\mathbf{t}^\mathrm{sw}$ the Cauchy stress of the subglacial water side of the singular surface, $\boldsymbol{v}^\mathrm{i}_\mathrm{b}$ the ice velocity and $\mathbf{t}^\mathrm{i}$ the stress field of the ice at the base.

We split the traction vector of the subglacial water in a normal and tangential component, with the water pressure $p^\mathrm{sw}$ and the stress in the normal direction. Following the same approach as at an ice shelf base (Greve and Blatter, 2009), we employ an empirical relation

$$\mathbf{t}^\mathrm{sw}\cdot\boldsymbol{n} = -p^\mathrm{sw}\boldsymbol{n} + C^\mathrm{i/sw}\rho^\mathrm{sw}|\boldsymbol{v}^\mathrm{sw}|^2\boldsymbol{e}_\mathrm{t} \tag{10}$$

with $\boldsymbol{e}_t = \boldsymbol{v}^{\mathrm{sw}}/|\boldsymbol{v}^{\mathrm{sw}}|$ and $\boldsymbol{e}_t \perp \boldsymbol{n}$. The drag coefficient at the underside of the ice is $C^{\mathrm{i/sw}}$, similar as a Manning roughness is taken into account in subglacial conduits. So that the part of the subglacial water becomes

$$\boldsymbol{v}^{\mathrm{sw}} \cdot \mathbf{t}^{\mathrm{sw}} \cdot \boldsymbol{n} = -p^{\mathrm{sw}} \boldsymbol{v}^{\mathrm{sw}} \cdot \boldsymbol{n} + \boldsymbol{v}^{\mathrm{sw}} \cdot C^{\mathrm{i/sw}} \rho^{\mathrm{sw}} |\boldsymbol{v}^{\mathrm{sw}}|^2 \boldsymbol{e}_t = -p^{\mathrm{sw}} v_{\perp}^{\mathrm{sw}} + C^{\mathrm{i/sw}} \rho^{\mathrm{sw}} |v_{\|}^{\mathrm{sw}}|^3 \tag{11}$$

with $v_{\perp}^{\mathrm{sw}}, v_{\|}^{\mathrm{sw}}$ the normal and tangential velocity of the subglacial water, respectively. This formulation is quite similar to the treatment of the jump condition at an ice shelf base. For the traction vector at the ice base, we follow the same procedure and find

$$\mathbf{t}^{\mathrm{i}} \cdot \boldsymbol{n} = -N\boldsymbol{n} + \tau_{\mathrm{b}} \boldsymbol{e}_t \tag{12}$$

with $N$ the normal component and $\tau_{\mathrm{b}}$ the component in the tangential plane. For $\boldsymbol{v}_{\mathrm{b}}^{\mathrm{i}} \cdot \mathbf{t}^{\mathrm{i}} \cdot \boldsymbol{n}$ we find

$$\boldsymbol{v}_{\mathrm{b}}^{\mathrm{i}} \cdot \mathbf{t}^{\mathrm{i}} \cdot \boldsymbol{n} = -N\boldsymbol{v}_{\mathrm{b}}^{\mathrm{i}} \cdot \boldsymbol{n} + \tau_{\mathrm{b}} \boldsymbol{v}_{\mathrm{b}}^{\mathrm{i}} \cdot \boldsymbol{e}_t. \tag{13}$$

With the jump of the internal energy $[\![u]\!] = L$, we can reformulate Eq. (8) to

$$q_{\perp}^{\mathrm{geo}} + q_{\perp}^{\mathrm{sw}} = \rho^{\mathrm{i}} a_{\mathrm{b}} L + \kappa(T) \operatorname{grad} T + p^{\mathrm{sw}} \boldsymbol{v}_{\perp}^{\mathrm{sw}} - C^{\mathrm{i/sw}} \rho^{\mathrm{sw}} |v_{\|}^{\mathrm{sw}}|^3 - N\boldsymbol{v}_{\mathrm{b}}^{\mathrm{i}} \cdot \boldsymbol{n} + \tau_{\mathrm{b}} \boldsymbol{v}_{\mathrm{b}}^{\mathrm{i}} \cdot \boldsymbol{e}_t. \tag{14}$$

The tangential components $C^{\mathrm{i/sw}} \rho^{\mathrm{sw}} |v_{\|}^{\mathrm{sw}}|^3$ and $\tau_{\mathrm{b}} \boldsymbol{v}_{\mathrm{b}}^{\mathrm{i}} \cdot \boldsymbol{e}_t$ are frictional heating and are dominating the contribution of heat arising from work of surface forces. They need to be seen as two end members of the system: either the ice is only in contact with a thick subglacial hydrological system, then $C^{\mathrm{i/sw}} \rho^{\mathrm{sw}} |v_{\|}^{\mathrm{sw}}|^3$ is active, or the subglacial hydrological system is permanently in contact with a lubricated base, then the second term $\tau_{\mathrm{b}} \boldsymbol{v}_{\mathrm{b}}^{\mathrm{i}} \cdot \boldsymbol{e}_t$ is governing. The components are visualized in Fig. 3.

Next, we aim at constraining the individual terms for which we use the following material parameters: $\rho^{\mathrm{i}} = 910\,\mathrm{kg\,m}^{-3}$, the latent heat of fusion, $L = 335\,\mathrm{kJ\,kg}^{-1}$, and the thermal conductivity for ice at the pressure melting point of $270.81\,\mathrm{K}$ $\kappa(270.81\,\mathrm{K}) = 2.10\,\mathrm{W\,m}^{-1}\,\mathrm{K}^{-1}$ (Greve and Blatter, 2009).

We consider three scenarios: (i) there is only temperate ice that is melting, (ii) heat is required to warm the ice to the pressure melting point and (iii) friction at the base is contributing significantly to basal melting. (i) For temperate ice and no heat arising from work of surface forces, we find a melt rates of at least $0.19\,\mathrm{m\,a}^{-1}$ to correspond to a heat flux of $1.84\,\mathrm{W\,m}^{-2}$. (ii) Considering $\operatorname{grad} T$ to be less than $10^{-1}\,\mathrm{K\,m}^{-1}$, this increases the required heat flux from scenario (i) by up to $0.21\,\mathrm{W\,m}^{-2}$, as this additional heat is required to warm the ice to the pressure melting point.

(iii) Heat arising from work of the surface forces may, however, reduce the required heat flux into the ice to melt this amount of ice. To this end, we need to estimate the magnitude of the components of the stress tensors.

We assume that the normal stress component $N$ is hydrostatic and bridging stresses to be negligible. With a mean density of ice of $910\,\mathrm{kg\,m}^{-3}$ we find $p^{\mathrm{i}} = 23.8\,\mathrm{MPa}$. The normal velocity is of the order of the basal melt rate $v_{\mathrm{b}}^{\perp} \approx -0.2\,\mathrm{m\,a}^{-1}$ by assuming the velocity of the interface ($\boldsymbol{w}$) to be zero. The normal component of the ice side is then in the order of $0.15\,\mathrm{W\,m}^{-2}$. For the tangential components of the ice side, we consider the shear stress at the base to be $\tau_{\mathrm{b}} \approx 1$ to $100\,\mathrm{kPa}$. This compares to basal shear stress found by Rückamp et al. (2020) of $50\,\mathrm{kPa}$. To constrain the sliding velocity, we assume it to be maximum the surface velocity of $57\,\mathrm{m\,a}^{-1}$ and minimum half of the surface velocity. This leads to a tangential component on the ice side to be up to $0.15\,\mathrm{W\,m}^{-2}$ (Fig. 4).

Next, we constrain the normal component of the subglacial water $p^{\mathrm{sw}}\boldsymbol{v}^{\mathrm{sw}}_{\perp}$. A water pressure of 10 to $23\,\mathrm{MPa}$ is consistent with subglacial hydrological modelling (Beyer et al., 2018; Smith-Johnsen et al., 2020a). Assuming the normal velocity to be

at most as large as the basal melt rate, we find the range of this term to be between 0.05 to $0.12\,\mathrm{W\,m^{-2}}$ (Fig. 4). The tangential component $C^{\mathrm{i/sw}}\rho^{\mathrm{w}}|\boldsymbol{v}^{\mathrm{sw}}_{\|}|^3$ needs an assumption on the roughness $C^{\mathrm{i/sw}}$, for which we consider a range from the roughness of the ice shelf base of $10^{-3}$ to a maximum roughness ten times as large.

The motivation for this is that ice shelf roughness is governed by convection cells at the interface, whereas in the inland ice, the interaction with the bedrock may lead to a larger roughness. As nothing is known about the shape of the subglacial conduit,

the range of velocity cannot be constrained well. We consider a speed similar to the one of the ocean $0.1\,\mathrm{m\,s^{-1}}$, but as surface rivers easily reach $1.0\,\mathrm{m\,s^{-1}}$, we take this as an upper limit (Fig. 4). Thus, the contribution of friction to the energy available for basal melting may account for at least $\sim 0.20\,\mathrm{W\,m^{-2}}$, with the potential to be far larger based on the assumptions we made.

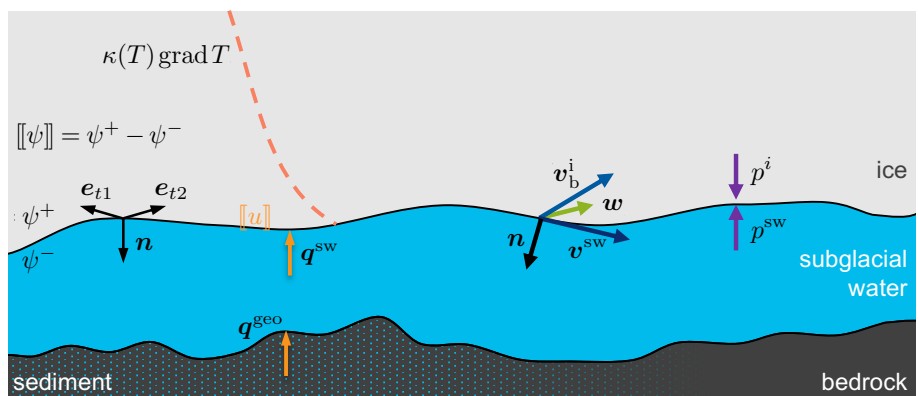

**Figure 3.** Sketch of the energy balance at the ice base. Thermal components are colored in orange, mechanical components are in blue-purple color. For detail explanation see main text.

To summarize, the jump in the tangential component (friction) has the potential to govern the heat budget, depending on

flow speeds in subglacial water and roughness of the ice base, as can be seen in Fig. 4. However, assuming the geothermal heat flux to be in the order of $\mathcal{O}(q^{\mathrm{geo}}) \approx 0.25\,\mathrm{W\,m^{-2}}$ makes evident that the key player in facilitating such high melt rates is the subglacial water system, that may supply the ice base with an additional heat flux.

We have focused our consideration onto the interface between a subglacial water layer and the ice, as this drives the basal melt rate. However, observations of Christianson et al. (2014) highlight the existence of a wet till layer beneath the ice stream.

Depending on the thickness of the water layer, the velocity and pressure of the water and the porosity of the till layer, complex interaction between the till and water may arise, too. Kutscher et al. (2019) present high resolution simulations of a comparable system that highlight the importance of studying this interface as well. To date, it is unclear which vertical extent of the water layer is required to decouple the interaction of a water-till interface from the ice-water interface and thus the influence on the basal melt rate.

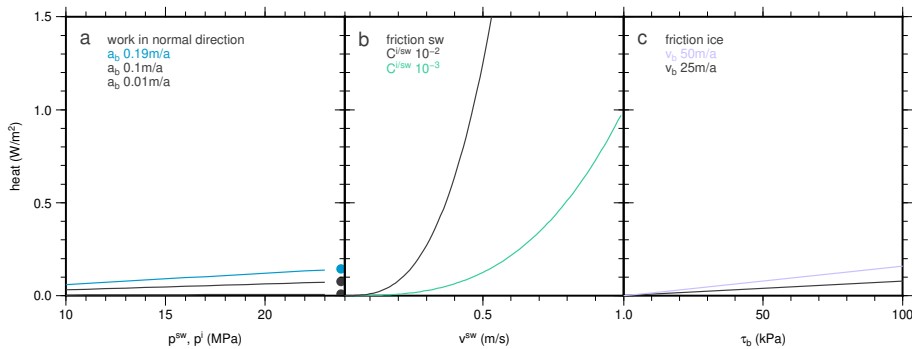

**Figure 4.** Magnitude of terms in the energy jump condition at the ice base. (a) Contribution of work of surface forces in normal direction from water pressure (lines) and ice overburden pressure (dots). (b) Contribution of work of surface forces in tangential direction from friction on the subglacial water. (c) Contribution of work of surface forces in tangential direction from friction on the ice side.

Large basal melting mainly affects basal sliding, as it increases the effective normal pressure. Considering sliding to be the dominant part of ice stream motion, large basal melt rates are also plausible in the respect that the subglacial hydrological system needs to be sustained over time and hence creep closure of cavities or conduits needs to be balanced by melting.

    Future measurements at EastGRIP after successful completion of the drilling to the ice base will shed more light onto the sliding speed and may also provide more information on the characteristics of the subglacial hydrological system. This will
enable the community to put our melt rate estimates into further context.

## 5   Conclusions

We estimated annual mean basal melt rates at the EastGRIP drill site from time series of high-precision phase-sensitive radar measurements. We derived the change in the measured ice thickness, thinning from firn densification occurring below the instrument and the vertical strain in the upper $1450\,\mathrm{m}$ of the roughly $2668\,\mathrm{m}$ thick ice. Two different scenarios for vertical
strain distribution were used to to quantify a plausible range of dynamic thinning. Thus, we derived an averaged melt rate of $0.19 \pm 0.04\,\mathrm{m\,a^{-1}}$. We are aware that these melt rates require an extremely large amount of heat that we suggest to arise from the subglacial water system and the geothermal heat flux. However, these melt rates are based on measurements with a modern ice penetrating radar whose penetration depth is limited due to transmitting power. Thus, no assumptions on past accumulation rates or other uncertainties in age reconstruction are involved. Our major uncertainty is the vertical strain in the lower part of the
ice stream. This could be overcome if a more powerful radar with a similar vertical resolution could be operated autonomously over several months.

*Code availability.*   Own developed MATLAB routines for ApRES processing are available from the corresponding author on request.

*Data availability.* Raw data of the ApRES measurements (https://doi.org/10.1594/PANGAEA.931018) are submitted to the World Data Center PANGAEA.

 **Appendix A: Additional figures of ApRES processing**

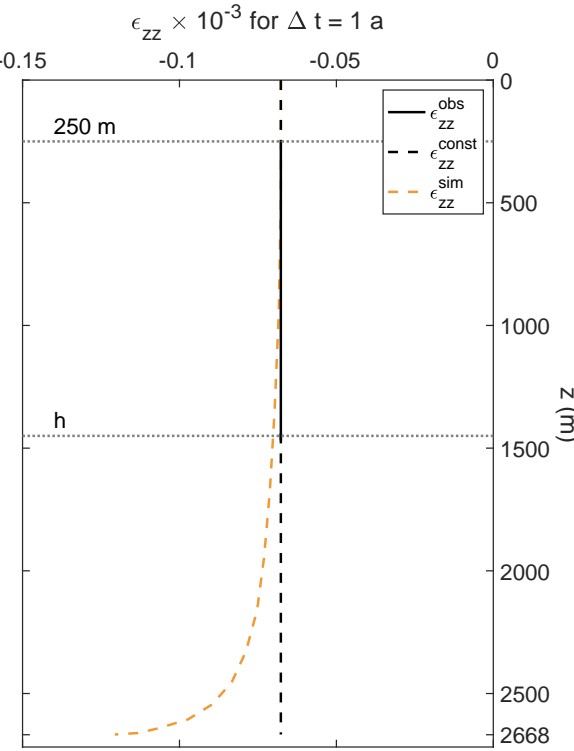

**Figure A1.** Observed vertical strain distribution (solid black line) and different extensions to the ice base (dashed lines) for two scenarios based on a constant (dashed black line), and a simulated (Rückamp et al., 2020, dashed orange line) distribution.

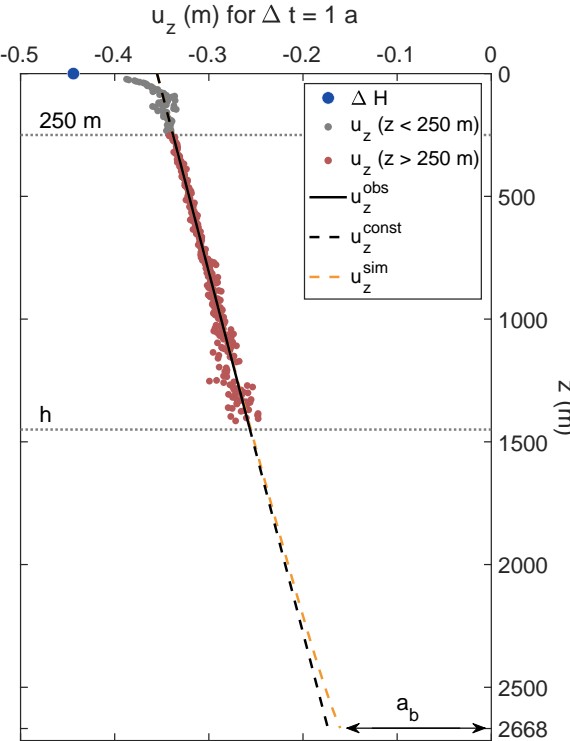

**Figure A2.** Derived vertical displacements $u_z$ of the depth of the ApRES ($\Delta H$; blue dot) and of selected internal layers referenced to the ice base from 2018/19 ApRES time series. Derived displacements used for melt rate estimations are marked by red dots and of layers within the firn by gray dots. The estimated displacements between a depth of $250\,\mathrm{m}$ and $h$ are used to calculate a linear fit (solid black line), the gradient of which is the vertical strain. Extrapolations to the bottom are shown by the dashed lines. The offset at the ice base is caused by basal melting and the difference between the intercept of the linear fit at $z = 0\,\mathrm{m}$ and $\Delta H$ is the firn compaction.

*Author contributions.* O.Z., A.H. conducted part of the field study. O.Z. processed the ApRES data and estimated the resulting basal melt rates. A.H. has designed the study and conducted the analysis of the energy jump condition. O.Z. and A.H. discussed the basal melt rate distribution and time series. O.Z. and A.H. wrote the manuscript jointly.

*Competing interests.* The authors declare that they have no conflict of interest.

*Acknowledgements.* Data has been acquired at the EastGRIP camp that kindly hosted this activity as an associate project. EastGRIP is directed and organized by the Centre for Ice and Climate at the Niels Bohr Institute, University of Copenhagen. It is supported by funding agencies and institutions in Denmark (A. P. Møller Foundation, University of Copenhagen), USA (US National Science Foundation, Office of Polar Programs), Germany (Alfred Wegener Institute, Helmholtz Centre for Polar and Marine Research), Japan (National Institute of Polar Research and Arctic Challenge for Sustainability), Norway (University of Bergen and Trond Mohn Foundation), Switzerland (Swiss National Science Foundation), France (French Polar Institute Paul-Emile Victor, Institute for Geosciences and Environmental research), Canada (University of Manitoba) and China (Chinese Academy of Sciences and Beijing Normal University). We like to thank Johanna Kerch, Sepp Kipfstuhl and Daniel Steinhage for their support in the deployment and the dismount of the measuring device. We are grateful for discussions with Thomas Kleiner and Martin Rückamp (AWI) and Ralf Greve (ITLS, Japan) on vertical strain-rates and our results.

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
