# Peer review of "Indication of high basal melting at the EastGRIP drill site on the Northeast Greenland Ice Stream"

_The Cryosphere, 2021_

## Referee Comment (RC1)

**Review of the article entitled "Indication of high basal melting at EastGRIP drill site on the Northeast Greenland Ice Stream"**

Basile de Fleurian

**1 General comments**

This study presents measurements of annual ice thickness changes at the EastGRIP site on the NEGIS and their implications regarding basal melt rates and heat fluxes at the base of the ice. The authors first present the measurement that were realised and the processing that was performed to obtain melt rate at the base of the ice. They then elaborate on the potential energy source at the base of the ice to give an overview of the different contribution as well as their relative importance.

I find this study very interesting and insightful on the relative importance of thermal processes at the base of the ice. The paper is well written and sufficiently supported by clear figures and method descriptions. I have a few general questions and comments bellow as well as some more specific points listed line by line.

Regarding the Dansgaard-Johnsen distribution of the vertical strain I feel that the paper is not clear on how this distribution was used. At first it is presented as a comparison point (Line 91) before being quickly discarded on the ground that it does imply a no-slide basal condition. I agree with the fact that this distribution of vertical strain is unlikely to appear in this setting but then it should probably not appear in the discussion and the conclusion should state only "two different scenarios" rather than three (Line 219).

The thickness variations you present for each year have a fairly small errors but the two different years that are presented actually show quite a large spread in the value. Your further analyse on those numbers and the different values that are computed with an alternative method (Line 133) leave me with the impression that the differences that we see here are more related to imprecision in the measurement rather than the natural variability of the thinning. Regarding that point, wouldn't it be more fair to consider the difference between these measure as uncertainty of the method rather than two different thinning rate for different years.

Finally I find the last part of the paper regarding the considerations on energy balance very interesting but I am missing a final summary of this section. It would be nice to

see all the heat sources summarised at the end with the range given by the different approximations. That would also shade more light on the possibility for NEGIS to have high melt rates with a reasonable geothermal heat flux.

**2   Specific comments**

Bellow is a list of more specific comments throughout the manuscript given with line numbers:

- Line 1: associated in place of "associate".

- Line 3: "is largely unknown".

- Line 4: Is "role" the proper term here, perhaps "relative importance" would fit better.

- Line 8: I understand that the value given here represent the different years, but wouldn't it be better to have a value of $0.19 \pm 0.04\,\mathrm{ma}^{-1}$.

- Line 14: My opinion here might be biased but I think that the work of Smith-Johnsen et al. (2020a) shows that we can model the NEGIS without relying on inversion.

- Line 15: Shouldn't it be "inability" here rather than "ability"?

- Line 20: "Increase in mass loss"?

- Line 22: I would say that one should aim at understanding the general dynamics and its different components rather than only lubrication.

- Line 23: "enhances" rather than "enable"?

- Line 24: In my opinion it is not the formation of the subglacial hydrological system that drive the sliding but more the increase in subglacial water pressure.

- Line 25: I would remove "system" here.

- Line 35: It should be stated here that this number is also tied to other parameters of the model. An other interesting point to touch upon might be the sensitivity of the NEGIS system to Geothermal Heat Flux as presented in Smith-Johnsen et al. (2020b).

- Line 73: A reference would be nice here to give a justification for those parameters.

- Line 87: The sentence starting on this line is hard to understand and could be rephrased.

- Line 97: "despite the fact that the no-slip boundary condition at the base is likely unrealistic..."

- Line 103: The notation for the vertical strain is not consistent throughout the manuscript, it is first introduced as $\Delta H_{\varepsilon_{zz}}$ but also appears as $\Delta H_{\epsilon_{zz}}$ and $\Delta H_{\epsilon}$

- Line 104: It should be stated clearly here that the Dansgaard Johnsen approximation have been discarded at this point.

- Line 111: $\Delta H_{\epsilon_{zz}}^{dj}$ here should be $\Delta H_{\varepsilon_{zz}}^{const}$

- Line 112: From the text I am not sure here what the surface refers to. Is it the topographic surface or the base of the ice cave on which the radar is set-up. I expect this is the latter but that should be clarified.

- Line 126: "as large as the one of..."

- Line 131: "Instead of comparing..."

- Line 139: "slightly" here seems like an understatement when the values you show are almost twice as large as the previous estimates.

- Line 141: "side" here should be "site".

- Line 172: Remove "are".

- Line 176: I would prefer "thick" here over "vertically extensive".

- Line 179: I wonder why the thermal conductivity for ice is taken at 273.15 K and not the pressure corrected melting point.

- Line 184: I am puzzled by scenario (ii). Which heat flux is raised? It seems that the raise is lower than the final heat flux or is there an issue with units?

- Line 191: The description of the bounds for the basal velocity here could be clearer.

- Line 193: The value here is expressed in milliwatts when all other values up to this point or in Watts, it might make it easier to read if the units were consistent. This also applies to lines 196 and 202.

- Line 197: Consider rephrasing as follows : "from the roughness of the ice shelf base to a maximum roughness ten time larger..."

- Line 200: Consider rephrasing as follows : "We consider a speed similar to the one of the ocean..."

- Line 203: Remove "by"

- Line 206: "we demonstrate" isn't the "we" missing?

- Figure A1: It would be nice to state the lines' colours to their scenarios in the caption of this figure.

**References**

Smith-Johnsen, S., de Fleurian, B., Schlegel, N., Seroussi, H., and Nisancioglu, K. (2020a). Exceptionally high heat flux needed to sustain the northeast greenland ice stream. *The Cryosphere*, 14(3):841–854.

Smith-Johnsen, S., Schlegel, N.-J., de Fleurian, B., and Nisancioglu, K. H. (2020b). Sensitivity of the northeast greenland ice stream to geothermal heat. *Journal of Geophysical Research: Earth Surface*, 125(1):e2019JF005252.

---

## Referee Comment (RC2)

**Review of 'Indication of high basal melting at EastGRIP drill site on the Northeast Greenland Ice Stream'**

March 19, 2021

**1 General comments**

This study presents estimated basal melt rates at the EastGRIP drill site obtained by ApRES measurements of two subsequent years. The manuscript is well structured and the writing is of high quality. The results presented are relevant and contribute to the ongoing scientific discussion about the driving mechanisms of ice flow and the role of the basal properties in the Northeast Greenland Ice Stream (NEGIS). The authors introduce the ApRES method and explain how the basal melt rates can be derived from repeated measurements, followed by a description of the processing steps applied to the ApRES data. The results are discussed and put into context through comparison to other studies and by discussing the energy balance at the base.

I think that the presented study provides data and results which are highly relevant and contribute to the current discussion about the basal state at the EastGRIP drill site. The presented basal melt rates are surprisingly high and might indicate how much the basal conditions of the NEGIS differs from its surrounding ice. I have some general comments and questions as well as specific feedback which hopefully is valuable to improve the quality of the paper.

The authors refer several times to earlier studies by MacGregor et al. (2016) and by Fahnestock et al. (2001) which derived basal melt rates of 0.1 ma$^{-1}$ or higher in the NEGIS. Both of these studies are based on radiostratigraphy methods using a 1D model of ice flow. While this type of model is valid for slow-moving areas, the local layer approximation (Waddington et al., 2007) in the NEGIS and its vicinity is not justified because the isochrone depths and shapes are considerably affected by ice flow dynamics. This is also emphasized by MacGregor et al. (2016), i.e. "...we restrict our interpretation of radiostratigraphy-inferred values of $\dot{m}$, h, and $\Phi$ to the portion of the GrIS where we consider the local layer approximation to be acceptable for reflections younger than 9 ka, i.e., the region where depth–age relationships may be represented reasonably by 1-D models that neglect horizontal gradients in ice flow". I consider it important to point out the restricted validity of these previous results in the NEGIS system when referring to the above-mentioned studies. The authors also point towards the study of Smith-Johnsen et al. (2020a), stating that they found a geothermal heat flux of 0.97 Wm$^{-2}$ to be necessary to reproduce the velocities of the NEGIS. While it is true that this result was obtained by Smith-Johnsen et al. (2020a) I think the context in which the reference is used here is misleading. Smith-Johnsen et al. (2020a) found such a high heat flux necessary to reproduce the NEGIS in *their model* with specific settings for basal parameters. However, they were also able to reproduce the ice stream with much lower basal heat flux in other studies (Smith-Johnsen et al., 2020b). From my point of

view, the introduction gives the reader the impression that basal melt rates of 0.1 ma$^{-1}$ and a geothermal heat flux of 0.97 Wm$^{-2}$ are likely in the NEGIS as these numbers were suggested by several previous studies. This is problematic because the fact that a heat flux of this order of magnitude exceeds the mean continental background by far (e.g. Alley et al., 2019) is neglected and the low probability as well as the restrictions of these previous results remain undiscussed (see e.g. Bons et al., 2020).

The basal melt rates in this study are derived from changes in the measured ice thickness, which is assumed to be a function of basal melt rate, vertical strain and firn densification. The authors thereby refer to similar studies by Nicholls et al. (2015), Vaňková et al. (2020) and Stewart et al. (2019) which use ApRES measurements to infer basal melt rates of ice shelves. A major difference between the application of ApRES on ice shelves and ice sheets is that the measured ice thickness on ice sheets is affected by the surface and bed topography as a result of ice flow, while the same measurement remains independent on lateral topography on floating ice shelves. In slow-moving areas of ice sheets, e.g. at ice domes, or if the method is applied over a short period of time, the effect of topography on the measurements might well be negligible. However, given the high ice-flow velocities and the distinctive bed and surface topography in the NEGIS, I am concerned about the fact that the impact of surface and bed slopes on the measured ice thicknesses at the EastGRIP drill site are not taken into account.

The evaluation of different scenarios for the vertical strain distribution are important to understand the sensitivity of the results towards the underlying assumptions. But I find it confusing that three scenarios are introduced but the results are only presented for two of them, since the Dansgaard-Johnsen model is discarded. I suggest to either include the results of the Dandgaard-Johnsen distribution in the manuscript or leave it out completely. Furthermore, as the vertical strain in the lower part of the ice column is considered the major uncertainty, it would make sense to me to provide the average between the different scenarios as result and consider the deviation from the mean as uncertainties. The errors provided in this manuscript seem very low as they include only the uncertainties of the measurements and might be misleading, as the total uncertainty of the inferred basal melt rates is clearly larger.

The evaluation of possible sources to provide the energy for the obtained basal melt rates is very interesting and an important aspect of the paper. However, I think that some essential elements are missing in the discussion. The suggested melt rates are larger than the present-day observed accumulation rates which has a considerable effect on the mass balance of the ice sheet. If such high melt rates were to persist over en extended period, I would expect to see evidence, e.g. in erosion of deep internal reflectors observed in radargrams. Radar images recorded in the vicinity of the study area do not show an extensive drag-down of internal layers compared to the surrounding (e.g. Keisling et al., 2014). It follows that the melt rates of the suggested order of magnitude must either be very local or a recent development. I believe that a more thorough discussion of these scenarios and the implications of the obtained results would add to the impact of the paper.

**2    Specific comments**

- Perhaps change the title to 'Indication of high basal melting at *the* EastGRIP drill site *in* the Northeast Greenland Ice Stream'?

- Line 1: change 'interior of the ice sheet' to 'interior of ice sheets' as it refers to ice streams in general

- Line 3: change 'are largely unknown' to 'is largely unknown' when referring to 'amount'

- Line 5: 'These findings' instead of 'these finding'

- Line 14: 'can only be reproduced well by such models if' instead of 'can only be represented well if'

- Line 15: should it be 'inability' instead of 'ability'?

- Line 20: perhaps '.. has already led to *ice flow* acceleration and *increased* mass loss?'

- Line 22: I would rather say that the general ice flow dynamics and its driving mechanisms are important to understand and not only the bed lubrication.

- Line 23: 'enable basal sliding *due to* a subglacial hydrological system' ?

- Line 25: just 'subglacial water' instead of 'subglacial water system'

- Line 28: Perhaps change this sentence to: 'The cause for such intensive melt was attributed to a high geothermal heat flux which possibly originates from the passage of Greenland over the Iceland hot spot'.

- Line 33: heat flux instead of heat fluxes.

- Line 77: It is not clear what the noise-level depth limit h is until the reader looks at Fig. 2. In the results it is stated that the vertical displacement can be estimated to a depth of 1450 m (assuming to be equivalent to h). Perhaps you can already write that h = 1450 m here.

- Line 59-79: I find the structure of this part a bit confusing. First you define $\Delta H_{\varepsilon_{zz}}$, then you describe all three quantities $\Delta H, \Delta H_f, \Delta H_{\varepsilon_{zz}}$ followed by a description of how the individual quantities are estimated. I suggest moving line 64-66 to the beginning of the paragraph.

- Line 66: You could also say here to what depth the densification processes are limited.

- Line 68: 'The vertical gradient of the vertical displacement is the vertical strain' seems a repetition of Eq.(4) and if so can be discarded.

- Line 76: the word 'measurement' is used four times in this sentence

- Line 77: should it not be 'time-consecutive measurement'? Also in line 131

- Line 82: measurement instead of measurements

- Line 91: 'scenarios to estimate $\Delta H_{\varepsilon_{zz}}$' instead of 'scenarios in order estimate a range $\Delta H_{\varepsilon_{zz}}$'

- Line 99: I'm confused by the way this is written. What is the reason behind assuming the shear flow onset being at the noise level? And why is overestimating the basal melt rate desirable?

- Line 101: It is not clear to me what is averaged here. If the measurement period extends only over 2 years I'd expect to get two mean annual values. Where do the 65 records come from?

- Line 106: Here you could refer to Fig. 2

- Line 152: $\rho$ and $\rho_i$ seem to be undefined.

- Line 153: Different notation in equation and the text e.g. $\mathbf{q}$ vs $q$ and $\omega$ vs $\mathbf{w}$

- Line 157: undefined term $\mathbf{v}^{SW}$ and $\mathbf{t}^{SW}$. 'SW' is used here as superscript while used as subscript in $q_{SW}$

**3 Technical comments**

- Section 2.3: inconsistent tenses, e.g. 'Firstly, we divided the depth profile' (line 79) vs 'Next, we estimate the vertical strain' (line 86)

- Figure 1: Is this red-green colormap suitable for readers with colourvision deficiencies?

- Inconsistent notation of $H_\epsilon$ and $H_\varepsilon$

- Perhaps consistently use either 'melt rate' or 'melting rate' throughout the text

- I believe that Tab. should be spelled out as Table, whereas Equation (3) should be abbreviated as Eq.(3).

- Figure 2: red and gray points are missing in the figure legend

- Heat fluxes are sometimes stated as $mWm^{-2}$ and sometimes as $Wm^{-2}$

**References**

Alley, R. B., Pollard, D., Parizek, B. R., Anandakrishnan, S., Pourpoint, M., Stevens, N. T., MacGregor, J. A., Christianson, K., Muto, A., and Holschuh, N. (2019). Possible Role for Tectonics in the Evolving Stability of the Greenland Ice Sheet. *Journal of Geophysical Research: Earth Surface*, 124(1):97–115.

Bons, P. D., de Riese, T., Franke, S., Llorens, M.-G., Sachau, T., Stoll, N., Weikusat, I., and Zhang, Y. (2020). Comment on "exceptionally high heat flux needed to sustain the northeast greenland ice stream" by s. smith-johnson et al., the cryosphere, 14, 841–854, 2020. *The Cryosphere Discussions*, 2020:1–5.

Fahnestock, M., Abdalati, W., Joughin, I., Brozena, J., and Gogineni, P. (2001). High geothermal heat flow, basal melt, and the origin of rapid ice flow in central Greenland. *Science*, 294(5550):2338–2342.

Keisling, B. A., Christianson, K., Alley, R. B., Peters, L. E., Christian, J. E., Anandakrishnan, S., Riverman, K. L., Muto, A., and Jacobel, R. W. (2014). Basal conditions and ice dynamics inferred from radar-derived internal stratigraphy of the northeast Greenland ice stream. *Annals of Glaciology*, 55(67):127–137.

MacGregor, J. A., Fahnestock, M. A., Catania, G. A., Aschwanden, A., Clow, G. D., Colgan, W. T., Gogineni, S. P., Morlighem, M., Nowicki, S. M., Paden, J. D., Price, S. F., and Seroussi, H. (2016). A synthesis of the basal thermal state of the Greenland Ice Sheet. *Journal of Geophysical Research: Earth Surface*, 121(7):1328–1350.

Nicholls, K. W., Corr, H. F., Stewart, C. L., Lok, L. B., Brennan, P. V., and Vaughan, D. G. (2015). A ground-based radar for measuring vertical strain rates and time-varying basal melt rates in ice sheets and shelves. *Journal of Glaciology*, 61(230):1079–1087.

Smith-Johnsen, S., De Fleurian, B., Schlegel, N., Seroussi, H., and Nisancioglu, K. (2020a). Exceptionally high heat flux needed to sustain the Northeast Greenland Ice Stream. *Cryosphere*, 14(3):841–854.

Smith-Johnsen, S., Schlegel, N. J., de Fleurian, B., and Nisancioglu, K. H. (2020b). Sensitivity of the Northeast Greenland Ice Stream to Geothermal Heat. *Journal of Geophysical Research: Earth Surface*, 125(1):1–14.

Stewart, C. L., Christoffersen, P., Nicholls, K. W., Williams, M. J., and Dowdeswell, J. A. (2019). Basal melting of ross ice shelf from solar heat absorption in an ice-front polynya. *Nature Geoscience*, 12(6):435–440.

Vaňková, I., Nicholls, K. W., Corr, H. F., Makinson, K., and Brennan, P. V. (2020). Observations of tidal melt and vertical strain at the filchner-ronne ice shelf, antarctica. *Journal of Geophysical Research: Earth Surface*, 125(1):e2019JF005280.

Waddington, E. D., Neumann, T. A., Koutnik, M. R., Marshall, H. P., and Morse, D. L. (2007). Inference of accumulation-rate patterns from deep layers in glaciers and ice sheets. *Journal of Glaciology*, 53(183):694–712.

---

## Author Comment (AC2)

**Authors point-to-point response on Referee Comment #1 to tc-2021-37**

**1. General Comments**

**#1**

*Regarding the Dansgaard-Johnsen distribution of the vertical strain I feel that the paper is not clear on how this distribution was used. At first it is presented as a comparison point (Line 91) before being quickly discarded on the ground that it does imply a no-slide basal condition. I agree with the fact that this distribution of vertical strain is unlikely to appear in this setting but then it should probably not appear in the discussion and the conclusion should state only "two different scenarios" rather than three (Line 219).*

Many thanks for raising this point. We agree that the use of the Dansgaard-Johnsen model has not been consistently applied and clearly described. We intended to demonstrate to people working more frequently with the DJ-type of profiles how this model would affect the basal melt rate. We follow your advice and that of Reviewer 2 and will remove this part entirely from the methods as well as from the results. We will keep a few sentences in the discussion explaining that a DJ-type of strain would lead to larger values for $a_b$, although the assumption the Dansgaard-Johnsen distribution is based on is rather unrealistic for an ice stream.

**Revised**:

*"A frequently used strain distribution (e.g., Fahnestock et al., 2001a; Keisling et al., 2014; MacGregor et al., 2016) that takes into account deviating strain within a shear zone is the Dansgaard–Johnsen distribution model (Dansgaard and Johnsen, 1969). As this model assumes a linearly decreasing strain in the shear zone that reaches zero at the ice base, the resulting basal melt rate at EastGRIP would be even larger. However, the Dansgaard–Johnsen model represents a no-slip boundary condition at the ice base. As this is an unrealistic assumption in an ice stream, we did not consider the Dansgaard–Johnsen model further."*

**#2**

*The thickness variations you present for each year have a fairly small errors but the two different years that are presented actually show quite a large spread in the value. Your further analyse on those numbers and the different values that are computed with an alternative method (Line 133) leave me with the impression that the differences that we see here are more related to imprecision in the measurement rather than the natural variability of the thinning. Regarding that point, wouldn't it be more fair to consider the difference between these measures as uncertainty of the method rather than two different thinning rates for different years.*

We agree that stating the average basal melt rate with the uncertainties based on the differences between both years gives a more realistic representation of the uncertainty of the method itself. We will update the stated melt rate to 0.19 ± 0.04 m/a. Many Thanks!

**#3**

*Finally, I find the last part of the paper regarding the considerations on energy balance very interesting but I am missing a final summary of this section. It would be nice to see all the heat sources summarised at the end with the range given by the different approximations. That would also shade more light on the possibility for NEGIS to have high melt rates with a reasonable geothermal heat flux.*

> We agree that a kind of table would be nice, but the reason to avoid such a table is, that some terms are 'active' when others are 'inactive'. The friction of subglacial water may become the dominant term when there is a thick water layer underneath, whereas the ice-side friction term is suppressed then. In addition, the velocity in the subglacial water system is indeed poorly constrained. Our approach so far was rather to take the perspective if it is reasonable to expect a substantial contribution of a particular term.

**2. Specific comments:**

Below is a list of more specific comments throughout the manuscript given with line numbers:

- *Line 1: associated in place of "associate".*

  > Agreed

- *Line 3: "is largely unknown".*

  > Agreed

- *Line 4: Is "role" the proper term here, perhaps "relative importance" would fit better.*

  > We agree with this point and will change *"role"* to *"relative importance"*.

- *Line 8: I understand that the value given here represent the different years, but wouldn't it be better to have a value of 0.19 ± 0.04 m/a.*

  > We fully agree with this point and will update the melt rate to 0.19 ± 0.04 m/a.

- *Line 14: My opinion here might be biased but I think that the work of Smith- Johnsen et al. (2020a) shows that we can model the NEGIS without relying on inversion.*

  > This is correct, we were focusing more on the benchmark experiment here, which involved spin-ups and inversion and both based on different

approximations, but no subglacial hydrological models. We will add a sentence to the manuscript about subglacial hydrological models.

**Version 1,** Line 14**:**

> *"The distinctive extent of Greenland's largest ice stream – the Northeast Greenland Ice Stream (NEGIS, Fig. 1) – can only be represented well if an higher-order approximation is considered for the momentum balance and initial states are based on inversion (Goelzer et al., 2018)."*

**Revised**:

> *"The distinctive extent of Greenland's largest ice stream – the Northeast Greenland Ice Stream (NEGIS, Fig. 1) – can only be reproduced well if a higher-order approximation is considered for the momentum balance and initial states are based on inversion (Goelzer et al., 2018)* **or involve subglacial hydrological models (Smith-Johnsen et al., 2020)***."*

● *Line 15: Shouldn't it be "inability" here rather than "ability"?*

Yes, "inability" is correct. Thanks!

● *Line 20: "Increase in mass loss"?*

Agreed

● *Line 22: I would say that one should aim at understanding the general dynamics and its different components rather than only lubrication.*

Thanks for raising this point. We agree and will change the sentence.

**Version 1,** Line 22**:**

> *"Consequently, it is expected and projected that NEGIS will contribute significantly to sea-level rise in the future (Khan et al., 2014), highlighting the importance to **understand its lubrication**."*

**Revised**:

> *"Consequently, it is expected and projected that NEGIS will contribute significantly to sea-level rise in the future (Khan et al., 2014), highlighting the importance to **understand the general ice flow dynamics and its driving mechanisms**."*

- *Line 23: "enhances" rather than "enable"?*

    We decided to use both: enables and enhances.

    **Version 1,** Line 23:

    *"One hypothesis for the genesis of NEGIS is locally increased basal melt rates at the onset area that **enable** basal sliding as basal melt water forms a subglacial hydrological system (Fahnestock et al., 2001a; Christianson et al., 2014; Franke et al.)."*

    **Revised**:

    *"One hypothesis for the genesis of NEGIS is locally increased basal melting at the onset area that **enables and enhances** basal sliding (Fahnestock et al., 2001a; Christianson et al., 2014; Franke et al., 2021) and forms a subglacial hydrological system."*

- *Line 24: In my opinion it is not the formation of the subglacial hydrological system that drive the sliding but more the increase in subglacial water pressure.*

    This is correct. We will rephrase it into two sentences, one referring to the publications that refer to the basal properties in the sense of detecting a wet base and discussing this to be a key player in formation of NEGIS, and a second sentence referring to the water pressure and publications that use/discuss this.

    **Version 1,** Line 23:

    *"One hypothesis for the genesis of NEGIS is locally increased basal melt rates at the onset area that enable basal sliding as basal melt water forms a subglacial hydrological system (Fahnestock et al., 2001a; Christianson et al., 2014; Franke et al.)."*

    **Revised**:

    "One hypothesis for the genesis of NEGIS is locally increased basal melting at the onset area that **enables and enhances** basal sliding (Fahnestock et al., 2001a; Christianson et al., 2014; Franke et al., 2021) and **forms a subglacial hydrological system. The coupling with basal sliding is facilitated via the water pressure, so that the sliding velocity rises with increasing water pressure (e.g., Beyer et al., 2018; Smith-Johnsen et al., 2020).**"

- *Line 25: I would remove "system" here.*

    Agreed

- *Line 35: It should be stated here that this number is also tied to other parameters of the model. Another interesting point to touch upon might be the sensitivity of the NEGIS system to Geothermal Heat Flux as presented in Smith-Johnsen et al. (2020b).*

    This is correct, the friction parameter also has a considerable impact on the dynamics of the ice stream.

    **Revised:**

    > *"By utilizing a coupled subglacial hydrology and ice sheet model, Smith-Johnsen et al. (2020b) demonstrated the large impact of an uncertainty in geothermal heat flux on the flow of NEGIS arising from the subglacial hydrological system, hence basal melting and water pressure, as well as from friction."*

- *Line 73: A reference would be nice here to give a justification for those parameters.*

    We will add **Fujita et al. (2000)**. There was a typo in the propagation velocity, which is 168,914 km/s, not 168,194 km/s.

- *Line 87: The sentence starting on this line is hard to understand and could be rephrased.*

    We will rephrase the sentence. Thanks!

    **Version 1,** Line 87**:**

    > *"We found a linear fit u_z(z) to match the curve of the cumulative vertical displacements of the remaining segments within the ice (below a depth of 250 m to exclude layers affected by firn densification) [equation]."*

    **Revised**:

    > *"To avoid influences of firn densification on the determination of ε^obs_zz, we excluded all segments above a depth of 250 m (~9 % of all segments). In addition, segments below the noise-level depth limit (depth at which the noise-level of the ApRES measurement prevents an unambiguous estimation) of h ≈ 1450 m were excluded (~45 % of all segments). Furthermore, outliers were filtered out (~7 %). We found a linear fit u_z(z) [equation] that best matches the cumulative vertical displacements of the remaining ~400 segments within the ice."*

- *Line 97: "despite the fact that the no-slip boundary condition at the base is likely unrealistic..."*

  Agreed

- *Line 103: The notation for the vertical strain is not consistent throughout the manuscript, it is first introduced as $\Delta H\_\varepsilon\_zz$ but also appears as $\Delta H\_\varepsilon\_zz$ and $\Delta H\_\varepsilon$*

  Thanks for pointing this out. We will change all "$\Delta H\_\varepsilon\_zz$" to "$\Delta H\_\varepsilon$".

- *Line 104: It should be stated clearly here that the Dansgaard Johnsen approximation have been discarded at this point.*

  We comment on this in the general comment #1.

- *Line 111: $\Delta H^{dj}$ here should be $\Delta H^{const}\_\varepsilon zz$*

  Agreed.

- *Line 112: From the text I am not sure here what the surface refers to. Is it the topographic surface or the base of the ice cave on which the radar is set-up. I expect this is the latter but that should be clarified.*

  Thanks for pointing this out. In this case *"surface"* meant position of the radar. We will change the sentence.

  **Version 1,** Line 112:

  > *"The firn densification – the intercept of the linear fit at the surface (see Fig. 2) – occurring below the radar is 0.074 m/a."*

  **Revised:**

  > *"The firn densification – the intercept (see Fig. 2) of the linear fit at z = 0 m (the elevation of the ApRES) – occurring below the radar is 0.074 m/a."*

- *Line 126: "as large as the one of..."*

  Agreed

- *Line 131: "Instead of comparing..."*

  Agreed

- *Line 139: "slightly" here seems like an understatement when the values you show are almost twice as large as the previous estimates.*

  We agree with this point and will remove *"slightly"*.

- *Line 141: "side" here should be "site".*

  Corrected.

- *Line 172: Remove "are".*

  Agreed.

- *Line 176: I would prefer "thick" here over "vertically extensive".*

  Yes, this is a good idea.

- *Line 179: I wonder why the thermal conductivity for ice is taken at 273.15 K and not the pressure corrected melting point.*

  Many thanks for raising this point. We fully agree that the conductivity should be calculated with the pressure corrected melting point. We calculated the thermal conductivity with the pressure corrected melting point and willupdate the computed values based on this number.

  **Version 1,** Line 179:

  > "Next, we aim at constraining the individual terms for which we use the following material parameters: $\rho i$ = 910 kg/m^3, the latent heat of fusion, L = 335 kJ/kg, and the thermal conductivity for ice at **273.15 K $\kappa$(273.15 K) = 2.07 W/(m K)** (Greve and Blatter, 2009)."

  **Revised**:

  > "Next, we aim at constraining the individual terms for which we use the following material parameters: $\rho^i$ = 910 kg/m^3, the latent heat of fusion, L = 335 kJ/kg, and the thermal conductivity for ice at **the pressure melting point of 270.81 K $\kappa$(270.81 K) = 2.10 W/(m K)** (Greve and Blatter, 2009)."

- *Line 184: I am puzzled by scenario (ii). Which heat flux is raised? It seems that the raise is lower than the final heat flux or is there an issue with units?*

  > With scenario (ii), we want to state a number by how much the required heat flux from scenario (i) would be increased if the interior ice is not tempered. We agree that this could be written more clearly. We additionally corrected a mistake regarding the unit of the temperature gradient and will remove the lower limit of 10^-3 K/m.

  > **Version 1,** Line 184:

  > > (ii) Considering grad T to be between 10^−1 and 10^−3 m/a raises the required heat flux into the ice by 0.207 to 0.0207 W/m^2 .

  > **Revised**:

  > > (ii) Considering grad T to be **less than 10^−1 K/m**, **this increases the required heat flux from scenario (i) by up to 0.21 W/m^2, as this additional heat is required to warm the ice to the pressure melting point.**

- *Line 191: The description of the bounds for the basal velocity here could be clearer.*

  > We will add the number of the surface velocity.

  > **Version 1,** Line 191:

  > > *"To constrain the sliding velocity we assume it to be maximum the surface velocity and minimum half of the surface velocity. "*

  > **Revised**:

  > > *"To constrain the sliding velocity, we assume it to be maximum the surface velocity **of 57 m/a** and minimum half of the surface velocity."*

- *Line 193: The value here is expressed in milliwatts when all other values up to this point or in Watts, it might make it easier to read if the units were consistent. This also applies to lines 196 and 202.*

  > Thanks for raising this point. We will change all units to Watts.

- *Line 197: Consider rephrasing as follows: "from the roughness of the ice shelf base to a maximum roughness ten time larger..."*

  > We will follow the suggestion and change the sentence.

- *Line 200: Consider rephrasing as follows: "We consider a speed similar to the one of the ocean..."*

    We will change the sentence as suggested.

- *Line 203: Remove "by"*

    Agreed.

- *Line 206: "we demonstrate" isn't the "we" missing?*

    We will rephrase this to 'makes evident' to avoid confusion.

- *Figure A1: It would be nice to state the lines' colours to their scenarios in the caption of this figure.*

    Agreed.

References

Beyer, S., Kleiner, T., Aizinger, V., Rückamp, M., and Humbert, A.: A confined–unconfined aquifer model for subglacial hydrology and its application to the Northeast Greenland Ice Stream, The Cryosphere, 12, 3931–3947, https://doi.org/10.5194/tc-12-3931-2018, 2018.

Christianson, K., Peters, L. E., Alley, R. B., Anandakrishnan, S., Jacobel, R. W., Riverman, K. L., Muto, A., and Keisling, B. A.: Dilatant till facilitates ice-stream flow in northeast Greenland, Earth and Planetary Science Letters, 401, 57–69, https://doi.org/10.1016/j.epsl.2014.05.060, 2014.

Dansgaard, W. and Johnsen, S.: A flow model and a time scale for the ice core from Camp Century, Greenland, Journal of Glaciology, 8, 215–223, https://doi.org/10.3189/S0022143000031208, 1969.

Fahnestock, M. A., Abdalati, W., Joughin, I., Brozena, J., and Gogineni, P.: High geothermal heat flow, basal melt, and the origin of rapid iceflow in central Greenland, Science, 294, 2338–2342, https://doi.org/10.1126/science.1065370, 2001a.

Franke, S., Jansen, D., Beyer, S., Neckel, N., Binder, T., Paden, J., and Eisen, O.: Complex Basal Conditions and Their Influence on Ice Flow at the Onset of the Northeast Greenland Ice Stream, Journal of Geophysical Research: Earth Surface, 126, e2020JF005689, https://doi.org/10.1029/2020JF005689, 2021.

Fujita, S., Matsuoka, T., Ishida, T., Matsuoka, K., and Mae, S.: A summary of the complex dielectric permittivity of ice in the megahertz range and its applications for radar sounding of polar ice sheets, in: Physics of ice core records, pp. 185–212, Hokkaido University Press, 2000.

Goelzer, H., Nowicki, S., Edwards, T., Beckley, M., Abe-Ouchi, A., Aschwanden, A., Calov, R., Gagliardini, O., Gillet-Chaulet, F., Golledge, N. R., et al.: Design and results of the ice sheet model initialisation initMIP-Greenland: An ISMIP6 intercomparison, The Cryosphere, 12, 1433–1460, https://doi.org/10.5194/tc-12-1433-2018, 2018.

Greve, R. and Blatter, H.: Dynamics of Ice Sheets and Glaciers, Springer Science & Business Media, https://doi.org/10.1007/978-3-642-03415-2, 2009.

Keisling, B. A., Christianson, K., Alley, R. B., Peters, L. E., Christian, J. E., Anandakrishnan, S., Riverman, K. L., Muto, A., and Jacobel, R. W.: Basal conditions and ice dynamics inferred from radar-derived internal stratigraphy of the northeast Greenland ice stream, Annals of Glaciology, 55, 127–137, https://doi.org/10.3189/2014AoG67A090, 2014.

Khan, S. A., Kjær, K. H., Bevis, M., Bamber, J. L., Wahr, J., Kjeldsen, K. K., Bjørk, A. A., Korsgaard, N. J., Stearns, L. A., Van Den Broeke, M. R., et al.: Sustained mass loss of the northeast Greenland ice sheet triggered by regional warming, Nature Climate Change, 4, 292–299, https://doi.org/10.1038/nclimate2161, 2014.

MacGregor, J. A., Fahnestock, M. A., Catania, G. A., Aschwanden, A., Clow, G. D., Colgan, W. T., Gogineni, S. P., Morlighem, M., Nowicki, S. M., Paden, J. D., et al.: A synthesis of the basal thermal state of the Greenland Ice Sheet, Journal of Geophysical Research: Earth Surface, 121, 1328–1350, https://doi.org/10.1002/2015JF003803, 2016.

Smith-Johnsen, S., de Fleurian, B., Schlegel, N., Seroussi, H., and Nisancioglu, K.: Exceptionally high heat flux needed to sustain theNortheast Greenland Ice Stream, The Cryosphere, 14, 841–854, https://doi.org/10.5194/tc-14-841-2020, https://tc.copernicus.org/articles/14/841/2020, 2020a.

Smith-Johnsen, S., Schlegel, N.-J., de Fleurian, B., and Nisancioglu, K. H.: Sensitivity of the Northeast Greenland Ice Stream to GeothermalHeat, Journal of Geophysical Research: Earth Surface, 125, e2019JF005 252, https://doi.org/doi.org/10.1029/2019JF005252, 2020b.

---

## Author Comment (AC3)

**Authors point-to-point response on Referee Comment #2 to tc-2021-37**

**1. General Comments**

**#1**

*The authors refer several times to earlier studies by MacGregor et al. (2016) and by Fahnestock et al. (2001) which derived basal melt rates of 0.1 m/a or higher in the NEGIS. Both of these studies are based on radiostratigraphy methods using a 1D model of ice flow. While this type of model is valid for slow-moving areas, the local layer approximation (Waddington et al., 2007) in the NEGIS and its vicinity is not justified because the isochrone depths and shapes are considerably affected by ice flow dynamics. This is also emphasized by MacGregor et al. (2016), i.e. "...we restrict our interpretation of radiostratigraphy-inferred values of m , h, and Φ to the portion of the GrIS where we consider the local layer approximation to be acceptable for reflections younger than 9 ka, i.e., the region where depth–age relationships may be represented reasonably by 1-D models that neglect horizontal gradients in ice flow". I consider it important to point out the restricted validity of these previous results in the NEGIS system when referring to the above-mentioned studies.*

> We understand the concern of the reviewer and will incorporate this into our manuscript. Basically, our measurements indicate that they are off by 90 % and melting is higher than this type of study suggests. Taking the perspective that the basic message of the radiostratigraphy method is to detect where considerable melting is taking place, they may, however, still do a very good job. Our observation confirms a high melt rate where the radiostratigraphy method suggested high melt rates, too.

> We want to comment that in future optimisation (e.g. inverse modelling) approaches may be able to constrain basal melt rates by optimising the mismatch between modelled and observed isochrones. The work that has been done by the radiostratigraphy community to develop the basis for such types of approaches has an enormous value. Its limitations today - and this is a very valid point of the reviewer - are not small, but this approach may in general to become the best strategy to survey ice sheet wide basal melt rates.

> **Version 1,** Line 25**:**

>> *"First estimates of basal melt rates by Fahnestock et al. (2001a) and later by MacGregor et al. (2016) are based on the interpretation of chronology in radiostratigraphy. Both studies found melt rates of 0.1 m/a and more – which is extremely large for inland ice."*

> **Revised:**

>> *"First estimates of basal melt rates by Fahnestock et al. (2001a) and later by* **Keisling et al. (2014) and** *MacGregor et al. (2016) are based on the interpretation of chronology in radiostratigraphy.* **All three studies** *found melt rates of 0.1 m/a and more – which is extremely large for inland ice.* **However,**

> *these estimates may be prone to limited validity given the assumptions about the flow regime and constant accumulation rate."*

**#2**

*The authors also point towards the study of Smith-Johnsen et al. (2020a), stating that they found a geothermal heat flux of 0.97 W/m^2 to be necessary to reproduce the velocities of the NEGIS. While it is true that this result was obtained by Smith-Johnsen et al. (2020a) I think the context in which the reference is used here is misleading. Smith-Johnsen et al. (2020a) found such a high heat flux necessary to reproduce the NEGIS in their model with specific settings for basal parameters. However, they were also able to reproduce the ice stream with much lower basal heat flux in other studies (Smith-Johnsen et al., 2020b). From my point of view, the introduction gives the reader the impression that basal melt rates of 0.1 m/a and a geothermal heat flux of 0.97 W/m^2 are likely in the NEGIS as these numbers were suggested by several previous studies. This is problematic because the fact that a heat flux of this order of magnitude exceeds the mean continental background by far (e.g. Alley et al., 2019) is neglected and the low probability as well as the restrictions of these previous results remain undiscussed (see e.g. Bons et al., 2020).*

Indeed, our intention is to inform the reader that a melt rate of 0.1 m/a has been estimated from previous observations and that a model also needed such a melt rate in order to simulate flow velocities similar to that of the NEGIS. However, a geothermal heat flux of 0.97 W/m^2 was required in the study from Smith-Johnsen et al. (2020a). Since Smith-Johnsen et al. (2020a) were aware that this amount of heat is too high to be explained by the geothermal heat flux alone, we will remove "geothermal" in the according sentence. Furthermore, we will add a sentence about the results presented in *Smith-Johnsen et al. (2020b).*

**Revised:**

> *"By utilizing a coupled subglacial hydrology and ice sheet model, Smith-Johnsen et al. (2020b) demonstrated the large impact of an uncertainty in geothermal heat flux on the flow of NEGIS arising from the subglacial hydrological system, hence basal melting and water pressure, as well as from friction."*

We have a detailed discussion in our manuscript highlighting how other sources of heat contribute to melting at the ice sheet base, including a discussion of geothermal heat flux versus heat flux from the subglacial hydrological system. All subglacial hydrology models so far (that we are aware of) and incapable of simulating a thermal regime. This is important to keep in mind when going into such details of assessing the geothermal heat flux of such approaches. A coupled ice-sheet-hydrology system is currently coupled via the water pressure that affects sliding. The message from studies like Smith-Johnsens is that a considerable amount of basal melt is needed to produce a certain amount of water supply into the hydrological system to build up a water pressure in a magnitude and distribution leading to ice stream flow as observed. What would be needed to solve this is a subglacial hydrological model that

resolves the water layer in the way that both, the velocity of the water, pressure AND temperature is computed. The lower boundary of that system would have a geothermal heat flux going into the system and provides a heat flux into the ice. This water system is unlikely laminar and turbulence may well play an important role. Now given that this needs to be done over a large area, this type of modelling is not around the corner. Due to a good reason why groundwater hydrologists are not running Navier-Stokes models on catchment scales. But even if we imagine due to enormous computing power in future to be feasible, we are still lacking the bed topography over that area. Radio echo sounding does 'only' give us the ice sheet base, not the bedrock topography. That would mean we would need seismics on the catchment scale (if seismics can resolve the thickness of the water layer adequately) or another method that does indeed survey the bedrock topography on catchment scale.

From our perspective, studies like Smith-Johnsen et al (2020a) should not be overly stressed on the geothermal heat flux that was needed to produce the velocity field of NEGIS, it is more about what melt rates are required to obtain a water pressure that is needed to sustain sliding leading to the observed surface velocities. This approach has its limitations, that is correct, but from our perspective more focus in the discussion should be on the basal melt rates and water pressure needed to produce NEGIS reasonably well, than the geothermal heat flux and/or friction parameter needed to produce these melt rates.

Smith-Johnsen et al. (2020b) shows the sensitivity of NEGIS to geothermal heat flux, but they did not show the velocity field and they did not state at all that they are able to reproduce the ice stream with a lower basal heat flux. Therefore, it remains unclear to us what the reviewer is referring to.

**Version 1,** Line 32**:**

> *"Smith-Johnsen et al. (2020) forced an ice model with a locally increased* **geothermal** *heat fluxes below the EastGRIP drill site and found that a heat flux of 0.97 W/m^2 (corresponding to a basal melt**ing** rate of 0.1 m/a (Fahnestock et al., 2001a)) is necessary to reasonably reproduce the velocities of NEGIS."*

**Revised:**

> *"Smith-Johnsen et al. (2020) forced an ice model with a locally increased heat flux below the EastGRIP drill site and found that a heat flux of 0.97 W/m^2 (corresponding to a basal melt rate of 0.1 m/a (Fahnestock et al., 2001a)) is necessary to reasonably reproduce the velocities of NEGIS."*

**#3**

*The basal melt rates in this study are derived from changes in the measured ice thickness, which is assumed to be a function of basal melt rate, vertical strain and firn densification. The authors thereby refer to similar studies by Nicholls et al. (2015), Vankova et al. (2020) and Stewart et al. (2019) which use ApRES measurements to infer basal melt rates of ice shelves. A major difference between the application of ApRES on ice shelves and ice sheets is that the measured ice thickness on ice sheets is affected by the surface and bed topography as a result of ice flow, while the same measurement remains independent on lateral topography on floating ice shelves. In slow-moving areas of ice sheets, e.g. at ice domes, or if the method is applied over a short period of time, the effect of topography on the measurements might well be negligible. However, given the high ice-flow velocities and the distinctive bed and surface topography in the NEGIS, I am concerned about the fact that the impact of surface and bed slopes on the measured ice thicknesses at the EastGRIP drill site are not taken into account.*

Many thanks for this point. We are happy to pick up this discussion and go here into some depth, as we have had such discussions a couple of times with community members already and it might be good to clarify a few points.

With ApRES measurements, we do measure the basal slope, but relative to the instrument and not as an absolute slope of the base. The measurement of the basal melt rate with ApRES is based on the ice thickness evolution equation. This equation is derived by vertically integrating the continuity equation over the vertical from the base to the surface, and the kinematic boundary conditions for the ice surface and the ice base. Kinematic boundary conditions are describing the motion of singular surfaces, such as the ice-atmosphere and ice-bedrock/hydrological system.

The resulting equation reads as

$$\frac{\partial H}{\partial t} = -\operatorname{div} \boldsymbol{Q} + a_s - a_b$$

with Q the volume flux, a_s the surface mass balance and a_b the basal melt rate. It is worth noting that this equation is independent of the configuration. It is the same for an Eulerian and Lagrangian reference system, hence the same in reference or present configuration. This equation means that an ice thickness change over time consists of stretching or compression, thus strain, at this particular site, and the accumulation/ablation at the surface and base.

Now let us consider a subglacial undulation. The ice will move over this protrusion by ascending and descending, so the ice base is exhibiting a slope. To some extent this will be accompanied by a slope at the surface. In all cases - with or without slope - the ice thickness H is the distance between the elevation of the ice surface h_s and ice base h_b: H = h_s - h_b. While crossing a bedrock undulation, the ice is deforming and it is this deformation of the ice plus the accumulation and ablation at the upper and lower surface, resulting in an ice thickness change.

This change in ice thickness may or may not come with a slope in surface and basal topography, which depends on the particular situation. A similar situation is given if ice passed over an area of higher water pressure $p_w$, leading to an increase in sliding velocity. This too will lead to a volume flux.

Now, as the volume flux equals for incompressible material the integral of the vertical strain over the ice column, a measurement of the vertical strain over depth is equal to a measurement of the volume flux over the ice column. This is what the concept of ApRES measurements is based on. Airborne surveys can in principle take the same strategy and redo the survey at exactly the same location a couple of times and derive the vertical strain, too. But a single airborne or ground based survey is lacking information of the vertical strain and needs to make assumptions on the volume flux and those estimates are then requiring slopes. All that is circumvented by (A)pRES measurements by directly measuring the vertical strain. The only disadvantage one may face with an (A)pRES system is that due to the low power of the transmitted radar burst, the absorption might be too large to measure strain down to the base, which requires then assumptions on the strain profile in that missing part. This is discussed in the manuscript in detail.

We hope that this extensive answer is shading some more light into that and helps to resolve the concerns. In the manuscript, we will add a paragraph starting from the ice thickness evolution equation, discussing all components and introducing the vertical strain rate.

**#4**

*The evaluation of different scenarios for the vertical strain distribution are important to understand the sensitivity of the results towards the underlying assumptions. But I find it confusing that three scenarios are introduced but the results are only presented for two of them, since the Dansgaard-Johnsen model is discarded. I suggest to either include the results of the Dandgaard-Johnsen distribution in the manuscript or leave it out completely.*

Many thanks for raising this point. We agree that the use of the Dansgaard-Johnsen model has not been consistently applied and clearly described. We intended to demonstrate to people working more frequently with the DJ-type of profiles how this model would affect the basal melt rate. We will follow your advice and remove this part entirely from the methods as well as from the results. We keep a few sentences in the discussion explaining that a DJ-type of strain would lead to larger values for $a_b$, although the assumption the Dansgaard-Johnsen distribution is based on is rather unrealistic for an ice stream.

**Revised**:

> *"A frequently used strain distribution (e.g., Fahnestock et al., 2001a; Keisling et al., 2014; MacGregor et al., 2016) that takes into account deviating strain within a shear zone is the Dansgaard–Johnsen distribution model (Dansgaard and Johnsen, 1969). As this model assumes a linearly decreasing strain in the shear zone that reaches zero at the ice base, the resulting basal melt rate at EastGRIP would be even larger. However, the Dansgaard–Johnsen model represents a no-slip boundary condition at the ice base. As this is an unrealistic assumption in an ice stream, we did not consider the Dansgaard– Johnsen model further."*

**#5**

*Furthermore, as the vertical strain in the lower part of the ice column is considered the major uncertainty, it would make sense to me to provide the average between the different scenarios as result and consider the deviation from the mean as uncertainties. The errors provided in this manuscript seem very low as they include only the uncertainties of the measurements and might be misleading, as the total uncertainty of the inferred basal melt rates is clearly larger.*

> We fully agree with this point and will state the melt rate for each year as suggested as an average value: 0.210 ± 0.015 m/a (2017/18) and 0.167 ± 0.018 m/a (2018/19). Many thanks!

**#6**

*The evaluation of possible sources to provide the energy for the obtained basal melt rates is very interesting and an important aspect of the paper. However, I think that some essential elements are missing in the discussion. The suggested melt rates are larger than the present-day observed accumulation rates which has a considerable effect on the mass balance of the ice sheet. If such high melt rates were to persist over an extended period, I would expect to see evidence, e.g. in erosion of deep internal reflectors observed in radargrams. Radar images recorded in the vicinity of the study area do not show an extensive drag-down of internal layers compared to the surrounding (e.g. Keisling et al., 2014). It follows that the melt rates of the suggested order of magnitude must either be very local or a recent development. I believe that a more thorough discussion of these scenarios and the implications of the obtained results would add to the impact of the paper.*

> We divide our answer to this point in three parts: (1) surface elevation change/thinning, (2) drag down of internal layers and (3) erosion of deep internal reflectors.
>
> 1. The mass balance is not driven by surface mass balance (accumulation) and basal melt rate alone. The change in mass balance the reviewer refers to, is an ice thickness change. As shown for point #3, the ice thickness evolution equation contains a term arising from the volume flux. If basal melt is larger than surface accumulation, the volume flux may still lead to no thickness change. A brilliant example for this is the 79N Glacier, where the floating tongue has basal melt rates in the order of tenths of meters per year and surface mass balance is negative, but still for a long period of time in the satellite observation there was no ice thickness change - the volume flux was large enough to sustain mass loss on the upper and lower surface. By no means the comparison of surface mass balance to basal mass balance alone is sufficient to conclude on ice thickness or surface elevation change.
> 2. Drag-down of internal layers is not solely due to variation in basal melt rates. Differences in sliding speeds and its feedback on viscosity, is changing internal layering as well (Fig. 4 in Leysinger Vieli et al., 2007, Fig. 6 in Gudlaugsson et al., 2016). No draw-down of layers may indicate basal melt

rates of similar magnitude or small gradient over the distance of the radar profile and/or no enhanced local sliding.

3. It is indeed an interesting question if such high melt rates are a recent phenomenon, or a variability in melt rates on time scales of millenia, centuries, decades and of course also the spatial variability is a very important question! With respect to terminology: every non-material singular surface experiences erosion, thus melting at the base, as well as erosion by dry friction is erosion of a layer. However, we think the reviewer means with erosion a disappearance of a deep layer along flow, a gradual drag-down with eventual disappearance. The scenarios under which this would happen are: (i) increasing melt rate along flow downstream, (ii) increasing sliding speed downstream plus constant melt rate and even (iii) increasing sliding speed and decreasing melt rate downstream, (iv) destruction of layers by increasing deformation downstream, (v) turbulent mixing of the lower layers appearing downstream. Scenarios under which no (gradual) erosion of layers are taking place are (a) constant melt rate, (b) constant sliding speeds, (c) freeze-on of subglacial water downstream. Consequently, there is no unique implication neither from erosion nor no erosion of layer to the magnitude of melt rates, nor its gradient along flow.

In addition, to our knowledge the lowermost part of the ice stream is not resolved well in the recent airborne campaigns to answer this. Maybe the new generation of radar that is currently developed will shade more light into that. Although it may be very difficult with a 'single' transect in time to distinguish between erosion and disappearance of a layer by destruction due to high shear strain at the base (the new generation of radar would also be most suitable to conduct a similar type of measurements as with the ApRES with a system with more power allowing to resolve layers further down and constraining the vertical strain further). So, we conclude, that there is neither evidence so far for NEGIS of erosion of layers nor no erosion of layers.

**2. Specific comments**

*Perhaps change the title to 'Indication of high basal melting at the EastGRIP drill site in the Northeast Greenland Ice Stream'?*

We appreciate your suggestion and are happy to take over "***the** EastGRIP drill site*" but prefer to stick with "***on** the Northeast Greenland Ice Stream*".

● *Line 1: change 'interior of the ice sheet' to 'interior of ice sheets' as it refers to ice streams in general*

Agreed

- *Line 3: change 'are largely unknown' to 'is largely unknown' when referring to 'amount'*

  Agreed

- *Line 5: 'These findings' instead of 'these finding'*

  Agreed

- *Line 14: 'can only be reproduced well by such models if' instead of 'can only be represented well if'*

  We will change the sentence as suggested

- *Line 15: should it be 'inability' instead of 'ability'?*

  Yes, "inability" is correct. Many thanks!

- *Line 20: perhaps '.. has already led to ice flow acceleration and increased mass loss?'*

  We will change the sentence as suggested.

- *Line 22: I would rather say that the general ice flow dynamics and its driving mechanisms are important to understand and not only the bed lubrication.*

  Thanks for raising this point. We agree and will change the sentence.

  **Version 1,** Line 22**:**

  *"Consequently, it is expected and projected that NEGIS will contribute significantly to sea-level rise in the future (Khan et al., 2014), highlighting the importance to **understand its lubrication.**"*

  **Revised**:

  *"Consequently, it is expected and projected that NEGIS will contribute significantly to sea-level rise in the future (Khan et al., 2014), highlighting the importance to **understand the general ice flow dynamics and its driving mechanisms.**"*

- *Line 23: 'enable basal sliding due to a subglacial hydrological system' ?*

  We will change this sentence this way, due to what is indeed stated in the cited papers and to address Reviewer 1's comment.

**Version 1,** Line 23:

> "One hypothesis for the genesis of NEGIS is locally increased basal melt rates at the onset area that enable basal sliding as basal melt water forms a subglacial hydrological system (Fahnestock et al., 2001a; Christianson et al., 2014; Franke et al.)."

**Revised**:

> "One hypothesis for the genesis of NEGIS is locally increased basal melting at the onset area that **enables and enhances** basal sliding (Fahnestock et al., 2001a; Christianson et al., 2014; Franke et al., 2021) **and forms a subglacial hydrological system**."

● *Line 25: just 'subglacial water' instead of 'subglacial water system'*

> Agreed

● *Line 28: Perhaps change this sentence to: 'The cause for such intensive melt was attributed to a high geothermal heat flux which possibly originates from the passage of Greenland over the Iceland hot spot'.*

> We agree and will change the sentence as suggested.

● *Line 33: heat flux instead of heat fluxes.*

> Agreed

● *Line 77: It is not clear what the noise-level depth limit h is until the reader looks at Fig. 2. In the results it is stated that the vertical displacement can be estimated to a depth of 1450 m (assuming to be equivalent to h). Perhaps you can already write that h = 1450 m here.*

> Thanks for raising this point. We agree with your suggestion and will add the depth.

> **Revised:**

> > "In addition, segments below the noise-level depth limit (depth at which the noise-level of the ApRES measurement prevents an unambiguous estimation) of **h ≈ 1450 m** were excluded (~45 % of all segments)."

- *Line 59-79: I find the structure of this part a bit confusing. First you define ΔHεzz , then you describe all three quantities ΔH, ΔHf,Δ Hεzz followed by a description of how the individual quantities are estimated. I suggest moving line 64-66 to the beginning of the paragraph.*

> As described in general comment #3, we will renew the introduction of this chapter and introduce the ice thickness evolution equation.

- *Line 66: You could also say here to what depth the densification processes are limited.*

> We will follow your suggestion and add the depth to which densification significantly affects the vertical displacement.

> **Revised:**

>> "Thus, ΔH is independent on the surface mass balance, $a\_s$ = 0 m/a, but influenced by firn densification **that significantly affects the vertical displacement in the upper ~100 m.**"

- *Line 68: 'The vertical gradient of the vertical displacement is the vertical strain' seems a repetition of Eq.(4) and if so can be discarded.*

> We will follow your suggestion and remove the sentence.

- *Line 76: the word 'measurement' is used four times in this sentence*

> We agree that the structure of the sentence could be improved. We will change the structure of this section slightly and mention the contrast to the method used by Vanková et al. (2020) a few sentences later.

> **Version 1,** Line 74**:**

>> "To derive vertical displacements of layers within the ice as well as for the basal return from the ApRES time series, we used a modification of the process described by Vanková et al. (2020). Both methods are based on estimated phase differences derived from cross-correlation of individual depth segments. **In contrast to Vanková et al. (2020), we compare the first measurement with each repeated measurement instead of pairwise time-consecutively measurements to reduce measurement errors**. Here, the ApRES time series is used to achieve a reliable estimation of the annual mean basal melt rate. First, we divided the depth profile into 6 m wide range segments with a 3 m overlap from a depth of 20 m below the antennas to 20 m above the ice base and a wider segment of 10 m (-9 to +1 m) around the basal return, characterized by a strong increase in amplitude. Each depth segment of the first measurement (t1) was

> cross-correlated with the same segment of each repeated
> measurements (ti)."

**Revised:**

> "To derive vertical displacements of layers and of the basal return
> from the ApRES time series, we modified the processing of Vanková
> et al. (2020) **to reduce measurement errors (details below)**. Both
> methods are based on phase differences estimated from cross-
> correlation of individual depth segments. Firstly, we divided the depth
> profile into 6 m segments with a 3 m overlap from a depth of 20 m
> below the antennas to 20 m above the ice base and a wider segment
> of 10 m (-9 to +1 m) around the basal return, characterized by a strong
> increase in amplitude. **In order to derive vertical displacements**,
> each depth segment of the first measurement ($t\_1$) was cross-
> correlated with the same segment of each repeated measurement
> ($t\_i$). **This is in contrast to Vanková et al. (2020), who derived
> displacements from pairwise time-consecutive measurements
> ($t\_(i-1) – t\_i$). "**

- *Line 77: should it not be 'time-consecutive measurement'? Also in line 131*

    We will change this in both lines as suggested.

- *Line 82: measurement instead of measurements*

    Agreed

- *Line 91: 'scenarios to estimate $\Delta H\varepsilon zz$ ' instead of 'scenarios in order estimate a range $\Delta H\varepsilon zz$'*

    We will change the sentence as suggested.

- *Line 99: I'm confused by the way this is written. What is the reason behind assuming the shear flow onset being at the noise level? And why is overestimating the basal melt rate desirable?*

    Our intention was to look at an extreme case: the shear flow would start at the point where we have no observations. If it would start further below, the effect on the basal melt rate would be lower. This is why we had chosen the 'end' of our observations as an onset of shear flow. The reviewer is entirely right in saying that this appears somewhat confusing. We will follow the reviewers general comment #4 and remove this part from the methods and the results.

    (Please note, this point was also raised by Reviewer 3 and is therefore also in that point2point answer)

- *Line 101: It is not clear to me what is averaged here. If the measurement period extends only over 2 years I'd expect to get two mean annual values. Where do the 65 records come from?*

    Yes, we agree, this was formulated rather confusingly.

    **Version 1,** Line 101**:**

    *"In order to be less dependent on a single measurement, we averaged the annual mean values of $\varepsilon\_obs$, $\Delta H\_\varepsilon\_zz$, $\Delta H\_f$ and $\Delta H$ from the last 65 records (roughly 25% of the measurements)."*

    **Revised:**

    *"In order to be less dependent on a single measurement, we compute for each of the last 65 days (records; roughly 25% of the measurements) of a year an annual melt rate and compute from these 65 melt rate estimates a mean annual value by averaging."*

- *Line 106: Here you could refer to Fig. 2*

    Agreed

- *Line 152: $\rho$ and $\rho i$ seem to be undefined.*

    Indeed! Both are the ice density and this has been corrected now. Many thanks!

- *Line 153: Different notation in equation and the text e.g. $\boldsymbol{q}$ vs $q$ and $\omega$ vs $\boldsymbol{w}$*

    We will correct this. Thanks!

- Line 157: undefined term $v^{sw}$ and $t^{sw}$ . 'sw' is used here as superscript while used as subscript in $q_{sw}$

    Thanks for pointing this out! We will correct this throughout the section.

**3. Technical comments**

- *Section 2.3: inconsistent tenses, e.g. 'Firstly, we divided the depth profile' (line 79) vs 'Next, we estimate the vertical strain' (line 86)*

    This is correct. We went through the entire text and hope that we found now all occasions of inconsistent tenses.

- *Figure 1: Is this red-green colormap suitable for readers with colourvision deficiencies?*

  We updated the colormap slightly and checked the new figure from someone with red-green color blindness.

- *Inconsistent notation of Hε and Hε*

  We will correct this.

- *Perhaps consistently use either 'melt rate' or 'melting rate' throughout the text*

  We will change "melting rate" to "melt rate".

- *I believe that Tab. should be spelled out as Table, whereas Equation (3) should be abbreviated as Eq.(3).*

  Thanks! We will change the spelling as suggested.

- *Figure 2: red and gray points are missing in the figure legend*

  We will add both to the legend.

- *Heat fluxes are sometimes stated as mW/m^2 and sometimes as W/m^2*

  Thanks for pointing this out! We changed all units to W/m^2.

**References**

Alley, R. B., Pollard, D., Parizek, B. R., Anandakrishnan, S., Pourpoint, M., Stevens, N. T., MacGregor, J. A., Christianson, K., Muto, A., and Holschuh, N.: Possible Role for Tectonics in the Evolving Stability of the Greenland Ice Sheet, Journal of Geophysical Research: Earth Surface, 124, 97–115, https://doi.org/10.1029/2018JF004714, 2019.

Bons, P. D., de Riese, T., Franke, S., Llorens, M.-G., Sachau, T., Stoll, N., Weikusat, I., and Zhang, Y.: Comment on "Exceptionally high heat flux needed to sustain the Northeast Greenland Ice Stream" by S. Smith-Johnson et al., The Cryosphere, 14, 841–854, 2020, TheCryosphere Discussions, 2020, 1–5, https://doi.org/10.5194/tc-2020-339, 2020.

Christianson, K., Peters, L. E., Alley, R. B., Anandakrishnan, S., Jacobel, R. W., Riverman, K. L., Muto, A., and Keisling, B. A.: Dilatant till facilitates ice-stream flow in northeast

Greenland, Earth and Planetary Science Letters, 401, 57–69, https://doi.org/10.1016/j.epsl.2014.05.060, 2014.

Dansgaard, W. and Johnsen, S.: A flow model and a time scale for the ice core from Camp Century, Greenland, Journal of Glaciology, 8, 215–223, https://doi.org/10.3189/S0022143000031208, 1969.

Fahnestock, M. A., Abdalati, W., Joughin, I., Brozena, J., and Gogineni, P.: High geothermal heat flow, basal melt, and the origin of rapid iceflow in central Greenland, Science, 294, 2338–2342, https://doi.org/10.1126/science.1065370, 2001a.

Franke, S., Jansen, D., Beyer, S., Neckel, N., Binder, T., Paden, J., and Eisen, O.: Complex Basal Conditions and Their Influence on Ice Flow at the Onset of the Northeast Greenland Ice Stream, Journal of Geophysical Research: Earth Surface, 126, e2020JF005689, https://doi.org/10.1029/2020JF005689, 2021.

Gudlaugsson, E., Humbert, A., Kleiner, T., Kohler, J., and Andreassen, K.: The influence of a model subglacial lake on ice dynamics and internal layering, The Cryosphere, 10, 751–760, https://doi.org/10.5194/tc-10-751-2016, 2016.

Keisling, B. A., Christianson, K., Alley, R. B., Peters, L. E., Christian, J. E., Anandakrishnan, S., Riverman, K. L., Muto, A., and Jacobel, R. W.: Basal conditions and ice dynamics inferred from radar-derived internal stratigraphy of the northeast Greenland ice stream, Annals of Glaciology, 55, 127–137, https://doi.org/10.3189/2014AoG67A090, 2014.

Khan, S. A., Kjær, K. H., Bevis, M., Bamber, J. L., Wahr, J., Kjeldsen, K. K., Bjørk, A. A., Korsgaard, N. J., Stearns, L. A., Van Den Broeke, M. R., et al.: Sustained mass loss of the northeast Greenland ice sheet triggered by regional warming, Nature Climate Change, 4, 292–299, https://doi.org/10.1038/nclimate2161, 2014.

Leysinger Vieli, G.-M., Hindmarsh, R., and Siegert, M.: Three-dimensional flow influences on radar layer stratigraphy, Annals of Glaciology, 46, 22–28, https://doi.org/10.3189/172756407782871729, 2007.

MacGregor, J. A., Fahnestock, M. A., Catania, G. A., Aschwanden, A., Clow, G. D., Colgan, W. T., Gogineni, S. P., Morlighem, M., Nowicki, S. M., Paden, J. D., et al.: A synthesis of the basal thermal state of the Greenland Ice Sheet, Journal of Geophysical Research: Earth Surface, 121, 1328–1350, https://doi.org/10.1002/2015JF003803, 2016.

Nicholls, K. W., Corr, H. F., Stewart, C. L., Lok, L. B., Brennan, P. V., and Vaughan, D. G.: A ground-based radar for mea-suring vertical strain rates and time-varying basal melt rates in ice sheets and shelves, Journal of Glaciology, 61, 1079–1087, https://doi.org/10.3189/2015JoG15J073, 2015.

Smith-Johnsen, S., de Fleurian, B., Schlegel, N., Seroussi, H., and Nisancioglu, K.: Exceptionally high heat flux needed to sustain theNortheast Greenland Ice Stream, The Cryosphere, 14, 841–854, https://doi.org/10.5194/tc-14-841-2020, https://tc.copernicus.org/articles/14/841/2020, 2020a.

Smith-Johnsen, S., Schlegel, N.-J., de Fleurian, B., and Nisancioglu, K. H.: Sensitivity of the Northeast Greenland Ice Stream to GeothermalHeat, Journal of Geophysical Research: Earth Surface, 125, e2019JF005 252, https://doi.org/doi.org/10.1029/2019JF005252, 2020b.

Stewart, C. L., Christoffersen, P., Nicholls, K. W., Williams, M. J., and Dowdeswell, J. A.: Basal melting of Ross Ice Shelf from solar heat absorption in an ice-front polynya, Nature Geoscience, 12, 435–440, https://doi.org/10.1038/s41561-019-0356-0, 2019.

Vanková, I., Nicholls, K. W., Corr, H. F., Makinson, K., and Brennan, P. V.: Observations of tidal melt and vertical strain at the Filchner-RonneIce Shelf, Antarctica, Journal of Geophysical Research: Earth Surface, 125, e2019JF005 280, https://doi.org/10.1029/2019JF005280, 2020.

Waddington, E. D., Neumann, T. A., Koutnik, M. R., Marshall, H. P., and Morse, D. L.: Inference of accumulation-rate patterns from deep layers in glaciers and ice sheets. Journal of Glaciology, 53(183): 694–712, 2007.

---

## Author Comment (AC4)

**Authors point-to-point response on Referee Comment #3 to tc-2021-37**

**1. General Comments**

**#1**

*The change in reconstructed basal melting from one year to the next seems to result mostly from differences in ΔH (measured) and ΔH_f (the offset from the linear fit to the phase-sensitive radar data). As mentioned in the text, whether ΔH is larger in '17/'18 than '18/'19 depends on the method used (compare Table 1 with lines 132–134). Thus, whether or not the basal melt rate was higher in one year or another comes down to ΔH_f, which to my understanding is the distance on the x-axis between the red dot at z=0 and the dotted line (Figure 2). If the authors want to make the claim that the basal melt rates in these years were indeed different (i.e. lines 115-116) they should provide more information about how robust their determination of ΔHf and in particular how the red dot at z=0 is defined and what the error on that measurement is, so the reader can be convinced that this difference is truly a robust indication that the system is somehow changing, principally due to firn densification, from one year to the next. Alternatively I think the results are equally robust and interesting if you consider the differences in reconstructed basal melt rate as indicative of the error in the method and provide one estimate of mean BMR based on 2 years of data.*

Many thanks for raising this point. The difference in basal melt rate from one year to the other is mainly caused by differences in the measured change of ice thickness ΔH. This change in ice thickness (the movement of the surface relative to the ice base) is shown by the red dot at z = 0 m in Fig. 2 and Appendix Fig. 2. As correctly mentioned by the reviewer, ΔH_f leads to slightly different numbers of ΔH. We are convinced that the estimation of ΔH is less robust than the estimation of ΔH_f since it is based on a displacement derived from only one segment.

However, we agree that stating the averaged basal melt rate with the uncertainties based on the differences between both years gives a more realistic representation of the uncertainty of the method itself. We will follow your suggestion and that of Reviewer 1 and update the stated melt rate to 0.19 ± 0.04 m/a instead of two separated melt rates.

**#2**

*How do the scenarios and assumptions about subglacial water flow relate to observation of a dilatant till layer beneath this site (Christianson et al. 2014)? I would like to see a discussion of this high-porosity, water-saturated till layer added to the discussion section. Wouldn't the presence of such a till layer promote more distributed subglacial flow, as opposed to the channelized flow assumed by the authors in for example lines 199-201?*

There are observations of combinations of Nye channels (incised into the sediment) and Röthlisberger channels (incised into the ice) existing, which demonstrates that a saturated till layer does not necessarily favor distributed flow or prevent channels of forming.

Indeed, the interaction between the porous till layer and the water layer is extremely interesting. There are case studies in which the flow of water in the porous medium and the water layer is simulated using the Navier-Stokes equation. In these types of studies that are coming with enormous computational costs, the porous medium is either approximated by a matrix of simple geometries (cubes, cylinders) or a CT-derived geometry is used. Figure 11 in Kutscher et al. (2019) is showing a situation which is likely very similar to the subglacial hydrological system with a wet till. Also Fig. 14 of the same publication shows nicely how strong the interaction between the flow in the channel and porous medium is in terms of velocity and pressure.

With respect to our choices of the values for the velocities in the water layer, we have no such direct simulation as Kutscher et al. (2019) for our system and no observation of the speed. Therefore, we have tried to take two end members, the speed in the ocean and of an open channel. If this is indeed capturing the maximum velocity well is unclear to date. It would be great if this could be measured when the EastGRIP is giving access to the bed.

**#3**
*There are relatively few places on earth where we have the active-source seismic measurements of Christianson et al. 2014 now coupled with these phase-sensitive radar observations and I think the authors have a very unique opportunity here to describe the processes and characteristics of this subglacial system in greater detail than they have already.*

We would be more than happy to obtain a better constraint or more knowledge on the subglacial hydrological system with the ApRES, but at the end, an ApRES does only survey the ice body and the response of the ice body to forcing at the ice base, may it arise from friction of a saturated till layer, from a thick water sheet or a channel. Only the 'ice side' is accessible with the ApRES.

**#4**
*In particular, I would also like to see further discussion of the velocity of the subglacial water system.*

We fully understand the intention of the reviewer and are ourselves interested in the subglacial water velocity. Currently, the subglacial hydrological models applied to NEGIS/EastGRIP are using an effective porous medium (EPM) layer approach and although this computes the flux from which the velocity can be constrained, the velocity in the porous medium may differ from the real world situation to some extent. To solve this, simulations resolving the water layer are required, so no porous medium approach anymore, but Navier-Stokes type of simulation for this water system. If that velocity matches the EPM derived velocity well, then we (the community) would be able to get more into the velocity of the water layer and this allows then to constrain by far better than we do here the frictional heat. But this does not only go beyond the scope of this paper, it is also not easy to achieve. We are however, still somewhat optimistic that a direct measurement of the water velocity may be possible when the EastGRIP consortium drills into that system.

Although we have now a relatively dataset over almost two years, this is still located at one spot. Our plans for the next field season are to deploy as many ApRES as possible with some distance to the EastGRIP camp and also one outside the main ice stream, to estimate the spatial variability of basal melt. Indeed, it would be best suited to match these locations with the seismic lines of Christianson et al. 2014.

**#5**

*Without any information about the shape of the conduit it is not possible to constrain the volume of water that would be required to maintain this heat flux into the subglacial system at NEGIS. Because there is no seasonal input of surface water (e.g., moulins) upstream of this study site, the authors' hypothesis requires a year-round steady source of subglacial water to maintain these basal melting rates in steady-state. Where do the authors think that water would originate? I would like to see further discussion on this topic.*

In our manuscript we present measurements of the basal melt rate and discuss which heat budget is needed to produce such melt rates.

It is correct that without the volume of the conduit, it is not possible to constrain the FLUX in the conduit and with that all contributions in the energy balance that contain the velocity of the water. In addition, the temperature field of the water is unknown and with that the heat flux arising from the water going into the ice $q^{sw}$ in our notation.

Indeed, a year-round, but not necessarily steady, source of subglacial water is likely to exist, although we only present a point-measurement at EastGRIP. Simulated basal melt rates are showing the area that experiences melt and airborne radar observations are used to infer a wet base, indicating melt, too. Some of the water in the water catchment will feed into the system at EastGRIP, as simulations of Smith-Johnsen et al., 2020a and Beyer et al., 2018 showed. Simulated basal melt rates might be off in magnitude to some extent, still the area experiencing melt may be relatively well constrained. In particular the study of Smith-Johnsen et al. 2020a showed where and which amount of basal melt rates is needed to produce a basal water pressure that leads to the ice stream in its present form. Given that recent studies of basal sliding laws are showing the appropriateness of the applied sliding laws (Maier et al., 2021), the implications of Smith-Johnsen et al.'s study must not be underestimated.

To conclude: there is a wide water catchment that supplies the subglacial hydrological system year-round.

**#6**

*The work of Karlsson and Dahl-Jensen (2015) may be interesting to engage with here as well, as their findings are highly relevant to this discussion.*

> Karlsson and Dahl-Jensen (2015) is only considering a routing scheme, which does not represent the hydrological system adequately. There are more approaches then EPM-type models (de Fleurian et al., 2014, Sommers et al., 2018, Beyer et al., 2018) that may also be well suited for the area around EastGRIP (e.g., GlaDS Werder et al. 2013, Hewitt 2011 type models), but an EPM model does represent both, efficient and inefficient drainage, hence sheet flow/distributed flow and channelised flow. Both types of models are, however, better suited for NEGIS from our perspective. Nevertheless, this goes beyond the scope of our manuscript, which only intends to present the measurement of basal melt rates.

**#7**

*In general, I find the discussion of the subglacial hydrological system very interesting and informative. I think this discussion would be further supported by a schematic figure which depicts the major processes and end-members that the authors consider in their arguments (i.e. lines 175–176). Otherwise I find it somewhat difficult to visualize the system that the authors are describing, which would help with evaluating the assumptions that they make in setting up their calculations and the strengths and shortcomings of those assumptions for describing the NEGIS system (see point 2).*

> Many thanks for raising this point! We fully agree and are happy to support our energy balance consideration with a schematic figure, that appears now as Fig. 3.

[Figure]

**2. Specific comments**

Additionally, I suggest the following minor edits and more specific questions:

- *Line 1 "associate" change to associated*

  Agreed

- *Line 24 "Franke et al." is missing a year*

  Thanks for the hint. The manuscript by Franke et al. has now been published, which is why we are now able to complete the citation information.

- *Line 27 Keisling et al. (2014) inferred spatially variable basal melt rates of 0.05–0.2 m/a for the same region from ground-based radar observations.*

  Many thanks for pointing us to this reference! We went through the manuscript and did not find the 0.2 m/a in the publication, but this statement for the central trunk. '*The average inferred basal-melt rate outside the ice stream is 0.05 m/a, which is significantly lower than the average basal-melt rate inside the ice stream (0.11 m/a) and in the line crossing the southeastern margin (0.09 m/a), but all are quite elevated*. Therefore, we will add the reference and keep the text with "*0.1 m/a and more*".

- *Line 30 Suggested phrasing: In order to directly observe, among other things, flow regimes and basal conditions…*

  We will change the sentence as suggested.

- *Line 79 "wide" is confusing here, I think the sentence functions equally well as "...we divided the depth profile into 6 m segments with a 3 m overlap..."*

  We will change the sentence as suggested.

- *Line 85-86 Why discard these segments? Is there any pattern in depth to which segments are discarded? What proportion of the data were discarded for this reason?*

  Yes, indeed we could have written this in more detail. We will enlarge this and give details of which and why segments are disregarded.

  **Version 1,** Line 85**:**

  *"Segments whose time series contain outliers or whose shift deviates significantly from their neighboring segments were discarded."*

**Revised:**

> "To avoid influences of firn densification on the determination of $\varepsilon\hat{}obs\_zz$, we excluded all segments above a depth of 250 m ($\sim$9 % of all segments). In addition, segments below the noise-level depth (depth at which the noise-level of the ApRES measurement prevents an unambiguous estimation) limit of h $\approx$ 1450 m were excluded ($\sim$45 % of all segments). Furthermore, outliers were filtered out ($\sim$7 %)."

● *Line 97 - I am not convinced by the argument that h (i.e. kink height) in the Dansgaard- Johnsen strain rate model can simply be assumed to be the depth limit of the radar instrument. I would like to see either some citations to motivate the choice of this depth as realistic for the kink height in the DJ model or a consideration of how uncertainty in the kink height affects the final estimates of BMR.*

Many thanks for raising this point. Indeed, there is no physical reason for assuming the kink to match the location of the depth limit. The only reason we have chosen this is that it would represent the upper limit. Our entire intention to discuss a DJ-type of profile was to help readers that are coming from the community applying DJ-models at other locations, like at drill locations on ice divides, into what would it mean to have made the DJ assumption in this particular case here.

As the way we used the DJ-model was obviously more confusing than helpful, we will follow the suggestion from Reviewer 2 and remove this part from the methods and the results. We will keep a few sentences in the discussion explaining that a DJ-type of strain would lead to larger values for $a_b$, although the assumption the Dansgaard-Johnsen distribution is based on is rather unrealistic for an ice stream.

**Revised**:

> *"A frequently used strain distribution (e.g., Fahnestock et al., 2001a; Keisling et al., 2014; MacGregor et al., 2016) that takes into account deviating strain within a shear zone is the Dansgaard–Johnsen distribution model (Dansgaard and Johnsen, 1969). As this model assumes a linearly decreasing strain in the shear zone that reaches zero at the ice base, the resulting basal melt rate at EastGRIP would be even larger. However, the Dansgaard–Johnsen model represents a no-slip boundary condition at the ice base. As this is an unrealistic assumption in an ice stream, we did not consider the Dansgaard–Johnsen model further."*

(Please note, this point was also raised by Reviewer 2 and is therefore also in that point2point answer)

- *Line 132-133 should read "time-consecutive measurements"*

    We will change the sentence as suggested.

- *Line 195 please provide citations following "...consistent with subglacial hydrological modelling," preferably those that share similar characteristics with your study site, e.g. little seasonal input of meltwater from upstream.*

    We will include references, both simulate the NEGIS without any seasonal water input, thus they have similar characteristics than the system we discuss here.

- *Lines 210-211 - Can you provide a back-of-the-envelope calculation for the creep closure rate for the kind of environment you are considering?*

    There is no way to infer the form of the channelised system, so width and thickness of the 'void' space and in fact, these are the critical quantities in doing such an estimation. A Master thesis (in German) was simulating closure rates for subglacial channels (T. Schultz, 'Viskoelastische Modellierung der Dynamik eines Gletschers als Antwort auf basales Schmelzen und die Oberflächenmassenbilanz', 2017 University of Bremen) taking a viscoelastic material model into account and conducting parameter sensitivity tests on width, thickness and water pressure. With a water pressure of 6 MPa the closure for a half-sphere-shaped channel to 5% of its original size takes in the order of 60 days. The water pressure has recently been simulated to be in the order of 20 MPa (Beyer et al., 2018, Smith-Johnsen et al., 2020a), which is by far larger.

- *Consider point #2 above - why would this system favor a channelized subglacial water system as opposed to distributed water flow within an actively deforming porous till layer (i.e. Christianson et al. 2014)?*

    It is yet to be determined by in-situ observation which system is underlying the ice stream. An actively deforming porous till layer may very well be part of this system, no doubts, but it won't be sufficient to transport that large amount of water, as a porous till layer is rather inefficient in terms of water transport. The amount of deformation in the till will hopefully be measured in 2022/2023, when the EastGRIP drill progresses to the base and hopefully a Ploughmeter (and/or other instruments) will shade more light into this.

- *Does the fact that the radar instrument was advecting along with the ice give you any information about the scale and extent of the subglacial channels you are hypothesizing, or are the subglacial channels just being advected along with the ice column?*

  > Channels that are incised into the ice, such as Röthlisberger channels, would be advected with the ice, but undergo transformation by changing water input (basal melt), melt-opening, creep-opening/closure over time. The radar instruments only measure the change in ice thickness over time, but - unfortunately - no thickness of the water layer.

- *Line 216 "high-precise" change to high-precision*
  Agreed

- *Figure 1. Legend - To me the legend should go the other way, with bigger numbers toward the top of the colorbar and smaller numbers at the bottom. Consider flipping the legend.*

  > We agree to this point and will change the legend of Fig. 1 accordingly.

- *Figure 2. What is the red dot at z=0, and how is it measured? In the caption, "which" change to "whose" or "... line), the gradient of which is the vertical..."*

  > Many thanks for pointing out that the red dot is not well described. The big dot at $z = 0$ m is the derived change in ice thickness $\Delta H$.

- *Figure 3. May be helpful to label the three panels a, b, and c. What are the three dots in the left-most panel and why do they not connect with the thin lines?*

  > Yes, indeed the panels are better referred to with a, b, c - we will change this. The three dots represent the ice overburden pressure $p^i$ and as the ice thickness is well known this is only one value, therefore a dot. The lines are representing the water pressure assumptions. These information are added to the figure caption.

- *Code availability: sentence should end "on request."*
  Agreed

- *Acknowledgements: "EGRIP" is used here instead of "EastGRIP" which is used in the title, main text and Figure 1. Should be the same everywhere.*

  > We changed EGRIP to EastGRIP as suggested. Many thanks!

**References used in this review**

Beyer, S., Kleiner, T., Aizinger, V., Rückamp, M., and Humbert, A.: A confined–unconfined aquifer model for subglacial hydrology and its application to the Northeast Greenland Ice Stream, The Cryosphere, 12, 3931–3947, https://doi.org/10.5194/tc-12-3931-2018, 2018.

Christianson, K., Peters, L. E., Alley, R. B., Anandakrishnan, S., Jacobel, R. W., Riverman, K. L., Muto, A., and Keisling, B. A.: Dilatant till facilitates ice-stream flow in northeast Greenland, Earth and Planetary Science Letters, 401, 57–69, https://doi.org/10.1016/j.epsl.2014.05.060, 2014.

Dansgaard, W. and Johnsen, S.: A flow model and a time scale for the ice core from Camp Century, Greenland, Journal of Glaciology, 8, 215–223, https://doi.org/10.3189/S0022143000031208, 1969.

de Fleurian, B., Gagliardini, O., Zwinger, T., Durand, G., Le Meur, E., Mair, D., and Råback, P.: A double continuum hydrological model for glacier applications, The Cryosphere, 8, 137–153, https://doi.org/10.5194/tc-8-137-2014, 2014.

Fahnestock, M. A., Abdalati, W., Joughin, I., Brozena, J., and Gogineni, P.: High geothermal heat flow, basal melt, and the origin of rapid iceflow in central Greenland, Science, 294, 2338–2342, https://doi.org/10.1126/science.1065370, 2001a.

Franke, S., Jansen, D., Beyer, S., Neckel, N., Binder, T., Paden, J., and Eisen, O.: Complex Basal Conditions and Their Influence on Ice Flow at the Onset of the Northeast Greenland Ice Stream, Journal of Geophysical Research: Earth Surface, 126, e2020JF005689, https://doi.org/10.1029/2020JF005689, 2021.

Hewitt, I. J.: Modelling distributed and channelized subglacial drainage: the spacing of channels, Journal of Glaciology, 57, 302–314, https://doi.org/10.3189/002214311796405951, 2011.

Karlsson, N. B. & Dahl-Jensen, D. Response of the large-scale subglacial drainage system of Northeast Greenland to surface elevation changes. *The Cryosphere* **9**, 1465–1479. 2015.

Keisling, B. A., Christianson, K., Alley, R. B., Peters, L. E., Christian, J. E., Anandakrishnan, S., Riverman, K. L., Muto, A., and Jacobel, R. W.: Basal conditions and ice dynamics inferred from radar-derived internal stratigraphy of the northeast Greenland ice stream, Annals of Glaciology, 55, 127–137, https://doi.org/10.3189/2014AoG67A090, 2014.

Kutscher, K., Geier, M., and Krafczyk, M.: Multiscale simulation of turbulent flow interacting with porous media based on a massively parallel implementation of the cumulant lattice Boltzmann method, Computers & Fluids, 193, 103 733,https://doi.org/10.1016/j.compfluid.2018.02.009, 2019

MacGregor, J. A., Fahnestock, M. A., Catania, G. A., Aschwanden, A., Clow, G. D., Colgan, W. T., Gogineni, S. P., Morlighem, M., Nowicki, S. M., Paden, J. D., et al.: A synthesis of the basal thermal state of the Greenland Ice Sheet, Journal of Geophysical Research: Earth Surface, 121, 1328–1350, https://doi.org/10.1002/2015JF003803, 2016.

Maier, N., Gimbert, F., Gillet-Chaulet, F., and Gilbert, A.: Basal traction mainly dictated by hard-bed physics over grounded regions ofGreenland, The Cryosphere, 15, 1435–1451, https://doi.org/10.5194/tc-15-1435-2021, 2021.

Smith-Johnsen, S., de Fleurian, B., Schlegel, N., Seroussi, H., and Nisancioglu, K.:
Exceptionally high heat flux needed to sustain theNortheast Greenland Ice Stream,
The Cryosphere, 14, 841–854, https://doi.org/10.5194/tc-14-841-2020,
https://tc.copernicus.org/articles/14/841/2020, 2020a.

Sommers, A., Rajaram, H., and Morlighem, M.: SHAKTI: Subglacial Hydrology and Kinetic,
Transient Interactions v1.0, Geoscientific Model Development, 11, 2955–2974,
https://doi.org/10.5194/gmd-11-2955-2018, 2018.

Werder, M. A., Hewitt, I. J., Schoof, C. G., and Flowers, G. E.: Modeling channelized and
distributed subglacial drainage in two dimensions,Journal of Geophysical Research:
Earth Surface, 118, 2140–2158, https://doi.org/10.1002/jgrf.20146, 2013

---

## Author Response (AR2)

Dear Editor,

Thank you for your input! We completely agree that Section 2.1 was by far too short, both as an individual section and too short to introduce the topic. We have now expanded this section by giving a brief introduction to the radar system as well as its scope and we listed details to the measurement at EastGRIP. Furthermore, we give a short overview of the course of a measurement as well as the processing to achieve an amplitude and phase profile. At the end of this section, we deal with the signal-to-noise ratio. In this way, we shifted the short description of the processing from Section 2.3 to 2.1 and focus on the melt rate analysis in 2.3.

We hope that this will make it easier for the reader to follow the logical order: 'What data?' (2.1) -> 'How to obtain melt rates from that type of measurement in general' (2.2) -> 'How did we derive the melt rate from our dataset in particular?' (2.3). This is why we suggest the structure as presented in the new version. However, we are open to change this. We tried to improve the titles of our subsection, which may also help to make this route more clear and to support the structure.

Please find attached the point-to-point response to your comments.
Again, many thanks for your support in improving this manuscript!

Best regards,
Ole and Angelika

L24: to understand => of understanding

**Agreed and changed accordingly**

L35: the passage of => the orientation of

**Reviewer 2 suggested 'passage' which we think fits quite well to describe a plate moving across a feature.**

L37: is drilled => is being drilled [or] has been drilled

**Changed to "is being drilled".**

L38: in the course of => as part of

**Agreed and changed accordingly**

L89-90: This second point does not seem immediately clear, and I believe it is related to the lack of information in Section 2.1. Over which range of ice thicknesses does the ApRES radar sounding give information? Ie, from which region of the ice column are you extrapolating? Why doesn't it measure down to the base? I would suggest this be made clear in Section 2.1, so that this statement more immediately understandable.

**Thanks for raising this point. Your are absolutely right. This sentence was misleading and could not be understood without the information given later in Section 2.3. Since these three points are also included in Section 2.3, we decided to remove them here.**
**The radar measured reflections down to the ice base. However, due to weak reflections in the lower part of the ice column, noise prevents a reliable estimation of displacement for this range in depth. This was stated in Section 2.3 and is now mentioned at the end of section 2.1. We have then shortened the corresponding sentence in Section 2.3.**

L90: "extrapolated to the surface (the location of the ApRES)" <= Do you mean "(from the location of the ApRES)"? I understand that the ApRES measurements are not taken at the surface, but ~8m down. This sentence as written seems to imply it is at the surface. I see this is explained in Sect. 2.3, which should therefore appear beforehand as suggested above.

**Thanks! Again, you are absolutely right. With "surface", we meant the location of the ApRES. As written above, the three points have been removed as they are explained in Section 2.3. Here and in other sentences, we have replaced "surface" with "depth of the ApRES".**

L142, now L147: cannot exclude that => cannot exclude the possibility that

**Agreed and changed accordingly**

L144, now L149: would be in average more => would be, on average, more

**Agreed and changed accordingly**

L200, now L204: is at place => is active

**Agreed and changed accordingly**

L223, now L226/227: "maximum roughness ten times larger 10^-3 to 10^-2." <= ten times larger than what? this statement is not clear. Please modify so that the reader can clearly see the values of $C^i_{sw}$ and $v_{sw}$ used.

**We changed it this way:**
**"[...], for which we consider a range from the roughness of the ice shelf base of 10^-3 to a maximum roughness ten times as large."**

L241, now L245: Sentence is unclear, please rephrase.

**We rephrased the sentence to:**
**"Large basal melting mainly affects basal sliding, as it increases the effective normal pressure."**

L245, now L248: "after successful completion" <= After successful completion of what?

**We changed this sentence as follows:**
**"[...] after successful completion of the drilling to the ice base […]"**

L258: "which we want to encourage herewith" <= Remove, not necessary.

**Agreed and changed accordingly**

---

## Author Response (AR3)

Dear Editor,

Thank you for your great feedback and the acceptance of the manuscript! We implemented all change requests as suggested.

We additionally changed the data availability statement from
"Raw data of the ApRES measurements (https://doi.org/10.1594/PANGAEA.931018) are **submitted to** the World Data Center PANGAEA."
to
"Raw data of the ApRES measurements (https://doi.org/10.1594/PANGAEA.931018) are **available at** the World Data Center PANGAEA.".

Best regards,
Ole and Angelika